# LongMemEval: Benchmarking Chat Assistants on Long-Term Interactive Memory

**Di Wu**[1]*, **Hongwei Wang**[2], **Wenhao Yu**[2], **Yuwei Zhang**[3]*, **Kai-Wei Chang**[1], **Dong Yu**[2]
[1]UCLA, [2]Tencent AI Lab Seattle, [3]UC San Diego
{diwu,kwchang}@cs.ucla.edu

## Abstract

Recent large language model (LLM)-driven chat assistant systems have integrated memory components to track user-assistant chat histories, enabling more accurate and personalized responses. However, their long-term memory capabilities in sustained interactions remain underexplored. We introduce LongMemEval, a comprehensive benchmark designed to evaluate five core long-term memory abilities of chat assistants: information extraction, multi-session reasoning, temporal reasoning, knowledge updates, and abstention. With 500 meticulously curated questions embedded within freely scalable user-assistant chat histories, LongMemEval presents a significant challenge to existing long-term memory systems, with commercial chat assistants and long-context LLMs showing a 30% accuracy drop on memorizing information across sustained interactions. We then present a unified framework that breaks down the long-term memory design into three stages: indexing, retrieval, and reading. Built upon key experimental insights, we propose several memory design optimizations including session decomposition for value granularity, fact-augmented key expansion for indexing, and time-aware query expansion for refining the search scope. Extensive experiments show that these optimizations greatly improve both memory recall and downstream question answering on LongMemEval. Overall, our study provides valuable resources and guidance for advancing the long-term memory capabilities of LLM-based chat assistants, paving the way toward more personalized and reliable conversational AI. Our benchmark and code are publicly available at https://github.com/xiaowu0162/LongMemEval.

## 1 Introduction

Large language models (LLMs) have exhibited impressive capabilities in solving diverse tasks through natural language, leading to numerous successful chat assistant applications (OpenAI, 2022; Microsoft, 2023). Nevertheless, LLMs face limitations on tasks relying heavily on personal knowledge accumulated through long-term user-AI interactions, such as psychological counseling or secretarial duties (Zhong et al., 2024). Failing to incorporate user background and preferences into responses can diminish the response's accuracy as well as user satisfaction. To personalize LLM-based assistants, long-term memory, the ability to memorize, recall, and reason with a long interaction history, is indispensable. Recently, several commercial (OpenAI, 2024; Coze, 2024) and open-source assistant systems with memory (Zhong et al., 2024; Zhang et al., 2024) have been introduced. These systems leverage techniques like compressing, indexing, and retrieving from chat histories to generate more accurate and personalized responses.

Despite these advances, there has been limited progress in holistically evaluating the memory capability in long-term interactions. While several benchmarks evaluate LLMs on understanding long chat histories (Xu et al., 2022a;b; Zhong et al., 2024; Maharana et al., 2024; Du et al., 2024; Kim et al., 2024), they have two major shortcomings. First, they do not accurately reflect user-AI interactions: many focus solely on human-human conversations (Xu et al., 2022a; Maharana et al., 2024; Kim et al., 2024), while others omit task-oriented dialogues, which represent a significant

---

*work done during internship at Tencent AI Lab, mentored by Hongwei and Wenhao.

portion of chat assistant usage and challenge memorization with the long-context inputs and long-form responses. Their interactive histories also typically have a non-configurable length spanning only a few thousand tokens, limiting the difficulty as current systems continue to improve. Second, current benchmarks' questions only offer a limited coverage of the memory abilities required in dynamic long-term interactions. For instance, MemoryBank (Zhong et al., 2024) and PerLTQA (Du et al., 2024) insufficiently evaluate the ability to synthesize information across numerous sessions or to reason with temporal metadata or time references. All long-term memory benchmarks including recent ones such as LoCoMo (Maharana et al., 2024) also fail to evaluate recall of information provided by the assistant or reasoning with updated user information.

We introduce LONGMEMEVAL, a comprehensive benchmark for assessing the long-term memory capabilities of chat assistants. LONGMEMEVAL consists of 500 manually created questions to test five core memory abilities: information extraction, multi-session reasoning, temporal reasoning, knowledge updates, and abstention. Each question requires recalling information hidden within one or more task-oriented dialogues between a user and an assistant. Inspired by the "needle-in-a-haystack" test (Kamradt, 2023), we design a pipeline to compile a coherent and length-configurable chat history for each question. A chat system, then, is required to parse the dynamic interactions online for memorization, and answer the question after all the interaction sessions. While the length of the history is freely extensible, we provide two standard settings for consistent comparison: LONGMEMEVAL$_S$ with approximately 115k tokens per problem and LONGMEMEVAL$_M$ with 500 sessions (around 1.5 million tokens). Preliminary evaluations highlight the difficulty of LONGMEMEVAL, as long-context LLMs show a 30%~60% performance drop on LONGMEMEVAL$_S$, and manual evaluations reveal that state-of-the-art commercial systems only achieved 30%~70% accuracy in a setting much simpler than LONGMEMEVAL$_S$ (§3.4).

Finally, we present a unified view for memory-augmented chat assistants. Leveraging LONG-MEMEVAL, we comprehensively analyze memory design choices across three key execution stages—indexing, retrieval, and reading—and four control points: value, key, query, and reading strategy. Experimental insights identify several effective memory designs:

- (§5.2) Instead of sessions, *round* is the more optimal granularity for storing and utilizing the interactive history. While further compression into individual user facts harms overall performance due to information loss, it improves the multi-session reasoning accuracy.

- (§5.3) While using a flat index with the memory values themselves as the keys is a strong baseline, further *expanding* the keys with extracted user facts improves both memory recall (9.4% higher recall@$k$) and downstream question answering (5.4% higher accuracy).

- (§5.4) Naive time-agnostic memory designs perform poorly on temporal reasoning questions. We propose a simple indexing and query expansion strategy to explicitly associate timestamps with facts and narrow down the search range, improving the memory recall for temporal reasoning by 6.8%~11.3% when a strong LLM is employed for query expansion.

- (§5.5) Even with perfect memory recall, accurately utilizing retrieved items is non-trivial. Applying Chain-of-Note (Yu et al., 2023) and structured data format (Yin et al., 2023) improves question answering accuracy by as much as 10 absolute points across three LLMs.

## 2 RELATED WORK

**Long-Term Dialogue Benchmarks** As the ability of dialogue systems improve, research starts to focus on long-term dialogue understanding beyond traditional dialogue modeling benchmarks (Budzianowski et al., 2018; Wei et al., 2018). Early works focused on language modeling evaluation on generating personalized responses from human-human (Xu et al., 2022a) or human-AI (Xu et al., 2022b) chat histories. To more precisely evaluate memory accuracy, subsequent benchmarks shifted toward question answering (QA, Reddy et al. (2019); Zhang & Choi (2021)). For example, MemoryBank (Zhong et al., 2024) features multi-day chat histories from 15 users with 194 human-written probing questions. LoCoMo (Maharana et al., 2024) includes 50 long-term chat histories and questions testing single-hop, multi-hop, temporal, commonsense, world knowledge, and adversarial reasoning. PerLTQA (Du et al., 2024) scales the evaluation to 3,409 dialogues and 8,593 questions, covering world knowledge, personal profiles, social relationships, events, and dialogue history. DialSim (Kim et al., 2024) evaluates models' memory ability by roleplaying TV show characters

Table 1: A comparison between LONGMEMEVAL and existing long-term memory benchmarks. We use different colors to denote human-human and human-AI dialogue. #Sess and #Q denote the total number of sessions and questions. Context depth is defined as the number of tokens in the history. Finally, we compare the coverage of five core abilities: information extraction (IE), multi-session reasoning (MR), knowledge update (KU), temporal reasoning (TR), and abstaining on unanswerable questions (ABS). *Not reported in the paper, based on our approximation. **at most 2 sessions.

| Benchmark | Domain | #Sess | #Q | Context Depth | Core Memory Abilities | | | | |
|---|---|---|---|---|---|---|---|---|---|
| | | | | | IE | MR | KU | TR | ABS |
| MSC (Xu et al., 2022a) | Open-Domain | 5k | - | 1k | ✗ | ✗ | ✗ | ✗ | ✗ |
| DuLeMon (Xu et al., 2022b) | Open-Domain | 30k | - | 1k | ✗ | ✗ | ✗ | ✗ | ✗ |
| MemoryBank (Zhong et al., 2024) | Personal | 300 | 194 | 5k | ✓ | ✗ | ✗ | ✓ | ✗ |
| PerLTQA (Du et al., 2024) | Personal | 4k | 8593 | 1M* | ✓ | ✗ | ✗ | ✗ | ✓ |
| LoCoMo (Maharana et al., 2024) | Personal | 1k | 7512 | 10k | ✓ | ✓ | ✗ | ✓ | ✓ |
| DialSim (Kim et al., 2024) | TV Shows | 1k–2k | 1M | 350k | ✓ | ✓** | ✗ | ✓ | ✓ |
| LONGMEMEVAL (this work) | Personal | 50k | 500 | 115k, 1.5M | ✓ | ✓ | ✓ | ✓ | ✓ |

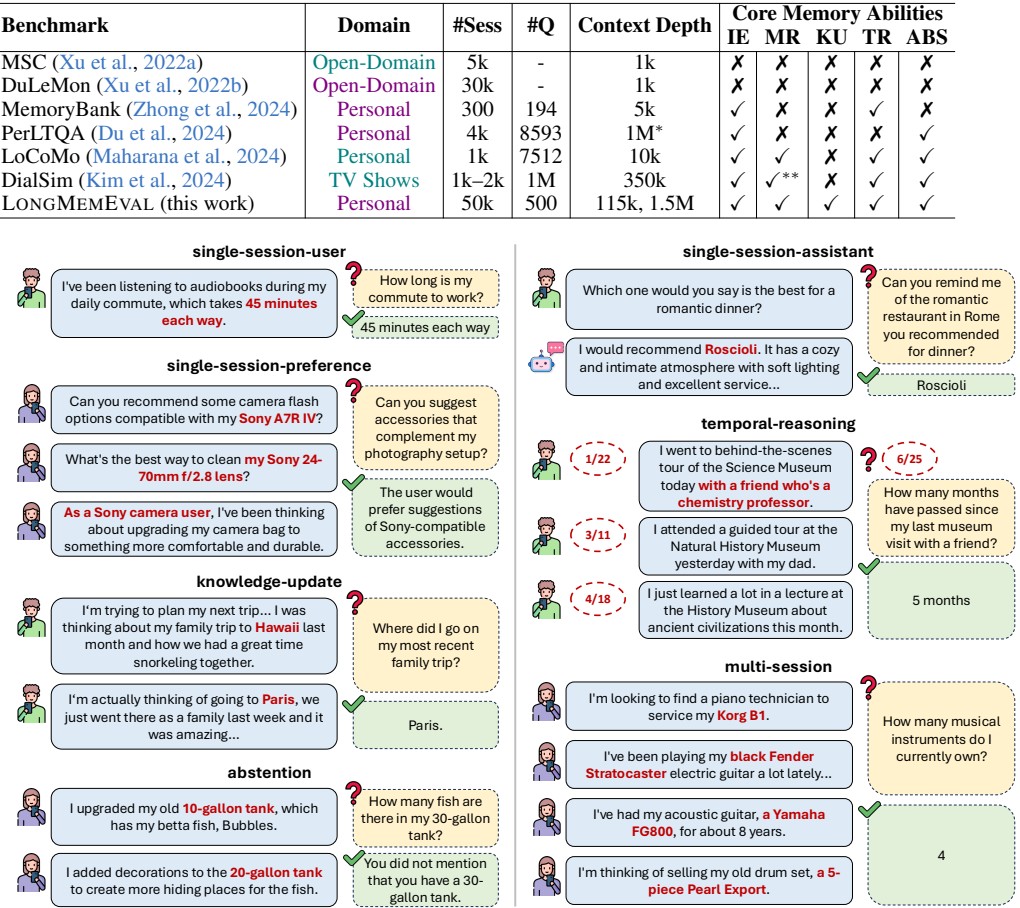

Figure 1: Examples of the seven question types in LONGMEMEVAL. For each example, we show the associated evidence statements on the left and the question with the answer on the right.

and introduces a time constraint that penalizes slow system responses. Despite these advancements, existing QA-based benchmarks overlook several memory capabilities critical to long-term user-assistant interactions: synthesizing information across numerous sessions, recalling assistant side information, and reasoning about updated user details or complex temporal references. Additionally, the chat histories are often too brief and do not reflect the nature of task-oriented interactions. Table 1 compares between LONGMEMEVAL and previous works, highlighting its advantages in both (1) featuring a long and freely extensible iterative history and (2) holistically covering critical memory abilities in a uniquely challenging way (further examples in Figure 1).

**Long-Term Memory Methods**  To equip chat assistants with long-term memory capabilities, three major techniques are commonly explored. The first approach involves directly adapting LLMs to process extensive history information as long-context inputs (Beltagy et al., 2020; Kitaev et al., 2020; Fu et al., 2024; An et al., 2024). While this method avoids the need for complex architectures, it is inefficient and susceptible to the "lost-in-the-middle" phenomenon, where the ability of LLMs to utilize contextual information weakens as the input length grows (Shi et al., 2023; Liu et al., 2024). A second line of research integrates differentiable memory modules into language models, proposing specialized architectural designs and training strategies to enhance memory capabilities (Weston et al., 2014; Wu et al., 2022; Zhong et al., 2022; Wang et al., 2023). Lastly, several studies approach long-term memory from the perspective of context compression, developing techniques

to condense lengthy histories into compact representations, whether in the form of LLM internal representations (Mu et al., 2023; Chevalier et al., 2023), discrete tokens (Jiang et al., 2023; Xu et al., 2024), or retrievable text segments via retrieval-augmented generation (RAG, Shi et al. (2024); Wang et al. (2023); Sarthi et al. (2024); Chen et al. (2023a); Gutiérrez et al. (2024)). Although LONGMEMEVAL can evaluate any memory system, we will take an online context compression perspective, where each history interaction session is sequentially processed, stored, and accessed on-demand through indexing and retrieval mechanisms (§4). This formulation aligns with current literature (Zhong et al., 2024; Gutiérrez et al., 2024) and commercial systems (OpenAI, 2024; Coze, 2024). Its plug-and-play nature also facilitates the integration into existing chat assistant systems.

## 3 LONGMEMEVAL

### 3.1 PROBLEM FORMULATION

The evaluation of LONGMEMEVAL requires an instance of 4-tuple $(\mathbf{S}, q, t_q, a)$. $\mathbf{S} \equiv [(t_1, S_1), (t_2, S_2), ..., (t_N, S_N)]$ is a sequence of $N$ history chat sessions ordered from the earliest to the latest, where $S_i$ is a multi-turn interaction between the user and a chat assistant and $t_i$ is the session's timestamp. Each session can be further decomposed into rounds: one user message followed by one assistant response. During test time, $\mathbf{S}$ is provided to the system one by one. $q$ and $t_q > t_N$ represent the question from the user and its date. $a$ is a short phrase indicating the answer, or a natural language rubric describing the preferred answer in the case where $q$ is open-ended.

### 3.2 LONGMEMEVAL: BENCHMARK CURATION

One major challenge in building a reliable personalized assistant is performing online recording, recalling, updating, and reasoning on the dynamically evolving user information. To comprehensively reflect the challenge, LONGMEMEVAL formulates five core long-term memory abilities:

- **Information Extraction (IE):** Ability to recall specific information from extensive interactive histories, including the details mentioned by either the user or the assistant.
- **Multi-Session Reasoning (MR):** Ability to synthesize the information across multiple history sessions to answer complex questions that involve aggregation and comparison.
- **Knowledge Updates (KU):** Ability to recognize the changes in the user's personal information and update the knowledge of the user dynamically over time.
- **Temporal Reasoning (TR):** Awareness of the temporal aspects of user information, including both explicit time mentions and timestamp metadata in the interactions.
- **Abstention (ABS):** Ability to identify questions seeking unknown information, i.e., information not mentioned by the user in the interaction history, and answer "I don't know".

As shown in Table 1, this formulation represents a more comprehensive ability coverage compared to prior long-term memory benchmarks like MemoryBank and PerLTQA. To thoroughly assess these abilities, LONGMEMEVAL features seven question types. **Single-session-user** and **single-session-assistant** test memorizing the information mentioned by user or assistant within a single session. **Single-session-preference** tests whether the model can utilize the user information to generate a personalized response. **multi-session** (MR) tests aggregating user information across two or more sessions. **knowledge-update** (KU) focuses on the ability to recognize changes in the user's life states and update the memory accordingly. **temporal-reasoning** (TR) tests reasoning with both the timestamp in metadata and explicit time references. Finally, we draw 30 questions from the previous question types and modify them into "false premise" questions, testing whether the model can correctly **abstain** from answering (ABS). Figure 1 presents an example for each question type.

**Question Curation** Figure 2 depicts the question curation pipeline. We define an ontology of 164 user attributes in five categories: lifestyle, belongings, life events, situations context, and demographic information. For each attribute, we leverage an LLM[1] to generate attribute-focused user background paragraphs, each of which includes detailed discussion of the user's life experience.

---

[1]Unless otherwise mentioned, Llama 3 70B Instruct (Dubey et al., 2024) is used as the LLM in the pipeline.

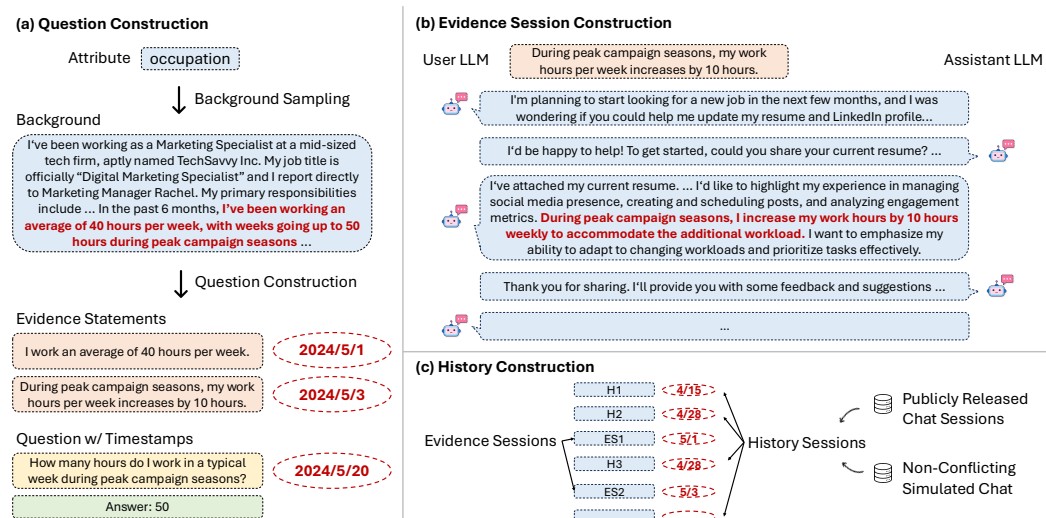

Figure 2: Data creation pipeline of LONGMEMEVAL. (a) Human experts construct all the questions and evidence statements. (b) Then, the evidence sessions are LLM-simulated and human-edited. (c) The full user-AI chat history is constructed at test time, whose length is freely configurable.

To create a question, we randomly sample a paragraph and use an LLM to propose several seed (question, answer) pairs. As these LLM-proposed questions often lack depth and diversity, human experts manually filter and rewrite all the questions to achieve the desired difficulty. Then, we manually decompose the answer into one or more *evidence statements* with optional timestamps.

**Evidence Session Construction** Each evidence statement is then separately embedded into a task-oriented *evidence session* created by self-chatting (Xu et al., 2023). The user LLM is instructed to convey the evidence statement indirectly, e.g., instead of stating "I bought a new car last month," it might instead ask for help about car insurance and reveal the information incidentally. This approach enhances the benchmark's difficulty by requiring systems to recognize and memorize user details not explicitly emphasized in conversations. We present the full details in Appendix A.1.

To ensure the data quality, all the evidence sessions are then manually screened and edited to (1) verify evidence inclusion, (2) distribute the evidence statement across different conversation positions, and (3) rephrase statements into more natural, colloquial language, especially for time mentions, which LLMs often express too formally. We also meticulously annotate the position of the evidence statement within each evidence session. For questions involving temporal information, we then manually add timestamps to both the evidence sessions and the questions. Most questions require evidence from multiple sessions (up to six) with evidence statements positioned diversely within sessions. Appendix A.3 presents further statistics of the final constructed data.

**History Compilation** For each question, LONGMEMEVAL compiles a coherent user-AI chat history (Figure 2c). Our approach is analogous to the needle-in-a-haystack test (Kamradt, 2023), which asks a model to retrieve brief information (the "needle") embedded in a long document (the "haystack"). In comparison, LONGMEMEVAL is more challenging and realistic as it involves retrieving and synthesizing information from multiple extended evidence sessions. Specifically, we sample a number of unrelated user-AI chat sessions, randomly insert the evidence sessions in the middle, and assign a plausible timestamp to all sessions. We draw the irrelevant sessions from two sources: (1) self-chat sessions simulated based on other non-conflicting attributes and (2) publicly released user-AI style chat data including ShareGPT (Zheng et al., 2023) and UltraChat (Ding et al., 2023). This design creates extensible realistic chat histories with minimal conflicts. While the pipeline allows us to compile chat histories of arbitrary length, we provide two standard settings: LONGMEMEVAL$_S$ (~115k tokens/question) and LONGMEMEVAL$_M$ (500 sessions, ~1.5M tokens).

## 3.3 EVALUATION METRIC

**Question Answering** As the correct answers can take flexible forms, an exact matching strategy as in previous works can result in inaccurate evaluations. To address this, LONGMEMEVAL

| System | LLM | Accuracy |
|---|---|---|
| Offline Reading | GPT-4o | 0.9184 |
| ChatGPT | GPT-4o | 0.5773 |
| | GPT-4o-mini | 0.7113 |
| Coze | GPT-4o | 0.3299 |
| | GPT-3.5-turbo | 0.2474 |

(a) Commercial memory-augmented chat assistants exhibit weak performance on LONGMEMEVAL. The accuracy of ChatGPT and Coze degrades by a large amount compared to directly reading the context ("Offline Reading") with the same LLM. Specifically, ChatGPT and Coze instantiated with GPT-4o exhibits 37% and 64% performance drop, respectively.

| Model | Size | Oracle | S | % Drop |
|---|---|---|---|---|
| **No Chain-of-Note** | | | | |
| GPT-4o | - | 0.870 | 0.606 | 30.3%↓ |
| Llama 3.1 Instruct | 70B | 0.744 | 0.334 | 55.1%↓ |
| Llama 3.1 Instruct | 8B | 0.710 | 0.454 | 36.1%↓ |
| Phi-3 128k Instruct | 14B | 0.702 | 0.380 | 45.9%↓ |
| Phi-3.5 Mini Instruct | 4B | 0.660 | 0.342 | 48.1%↓ |
| **With Chain-of-Note** | | | | |
| GPT-4o | - | 0.924 | 0.640 | 30.7%↓ |
| Llama 3.1 Instruct | 70B | 0.848 | 0.286 | 66.3%↓ |
| Llama 3.1 Instruct | 8B | 0.710 | 0.420 | 40.8%↓ |
| Phi-3 128k Instruct | 14B | 0.722 | 0.344 | 52.4%↓ |
| Phi-3.5 Mini Instruct | 4B | 0.652 | 0.324 | 50.3%↓ |

(b) Long-context LLMs exhibit large QA performance drops on LONGMEMEVAL$_S$ (column "S"), compared to the accuracy of answering the questions based on only the evidence sessions (column "Oracle").

Figure 3: Pilot study of (a) commercial systems and (b) long-context LLMs on LONGMEMEVAL.

employs a LLM to assess response quality (Liu et al., 2023). Specifically, we prompt-engineer the `gpt-4o-2024-08-06` model via the OpenAI API. Our meta-evaluation study demonstrates that the evaluator achieves more than 97% agreement with human experts. The prompts for each problem type as well as the human meta-evaluation details are presented in Appendix A.4.

**Memory Recall** As LONGMEMEVAL contains human-annotated answer location labels, intermediate retrieval metrics can be easily calculated if the chat system exposes its retrieval results. We report Recall@$k$ and NDCG@$k$, where $k$ is the number of top items retrieved by the system.

### 3.4 LONGMEMEVAL REPRESENTS A SIGNIFICANT CHALLENGE

Using LONGMEMEVAL, we conduct a pilot study on commercial systems and long-context LLMs.

**Commercial systems** We evaluate two commercial systems that maintain a set of memorized user facts as the user chats with the assistant: ChatGPT (OpenAI, 2024) and Coze (Coze, 2024). Since these systems only support memory features via their web interfaces, we randomly selected 97 questions and created a short chat history of 3-6 sessions (approximately 10x shorter than LONGMEMEVAL$_S$). Human annotators interacted with the chat assistants session-by-session and turn-by-turn, and finally ask the question in a new session[2]. In Figure 3a, we compare this online memory evaluation with offline reading, where GPT-4o is prompted to answer with the complete history provided as context. Both ChatGPT and Coze exhibited significant performance drops compared to offline reading, underscoring the challenging nature of LONGMEMEVAL. We found ChatGPT tended to overwrite crucial information as the chat continues, while Coze often failed to record indirectly provided user information. We provide analyses in Appendix B. Overall, this result highlights the **gap between building a seemingly personalized chat assistant by recalling isolated facts and demonstrating a genuinely strong memory ability**.

**Long-Context LLMs** While LONGMEMEVAL poses a significant challenge to online memory systems, is the benchmark easily tackled with offline reading over the entire history? In Figure 3b, we evaluated four advanced long-context LLMs on LONGMEMEVAL$_S$ (with a history length of approximately 115k tokens): GPT-4o, Llama 3.1 Instruct (Dubey et al., 2024), and Phi-3 (Abdin et al., 2024a). Compared to the oracle retrieval setting (answering with only the evidence sessions as the context), these LLMs showed a 30% to 60% performance decline when tasked with reading the entire LONGMEMEVAL$_S$ history, regardless of whether the chain-of-note technique (Yu et al., 2023) was applied (discussed further in §4.2). As the histories in LONGMEMEVAL$_S$ is still short (∼50 sessions), the performance is likely to further degrade as the interaction history expands. Overall, these findings suggest that **even the most capable current long-context LLMs struggle to manage an ever-growing interaction history without an effective memory mechanism**.

---

[2]All evaluations were conducted in the first two weeks of August 2024.

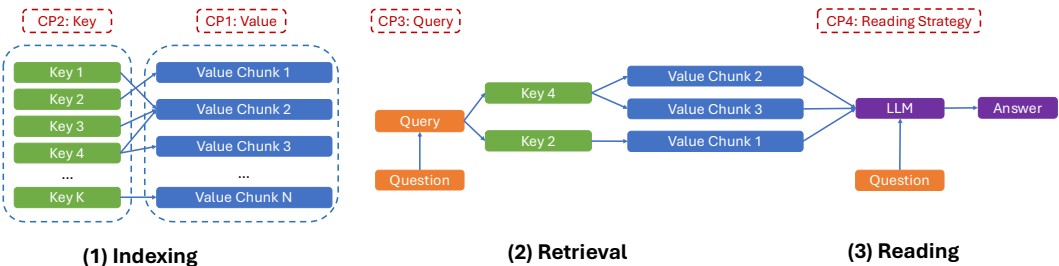

Figure 4: A unified view of a chat assistant with long-term memory in operation. We formulate three stages and four control points (CP). We provide further examples in Table 2 and Appendix C.

## 4 A UNIFIED VIEW OF LONG-TERM MEMORY ASSISTANTS

In this section, we formulate a three-stage long-term memory model for chat assistants. Despite its simplicity, this model provides a unified view of existing long-term memory assistant works. Along each of its stages, we then investigate crucial control points and propose our optimizations.

### 4.1 LONG-TERM MEMORY SYSTEM: FORMULATION

We formulate long-term memory as a massive key-value datastore $[(k_1, v_1), (k_2, v_2), ...]$. The keys $k_i$ can be heterogeneous, and could be discrete or continuous. In the discrete case, the key could be a sentence, a paragraph, a fact, or an entity, etc. In the continuous case, the key could be e.g., the model's internal representation under some inputs. The values $v_i$ might repeat. As shown in Figure 4, we formulate three stages for a memory-augmented assistant: (1) *indexing*, converting each history session $(t_i, S_i)$ into one or more key-value items, (2) *retrieval*, formulating a retrieval query and collecting $k$ most relevant items, and (3) *reading*, an LLM $\mathcal{M}$ reads the retrieval result and generates a response. In Table 2, we show how nine memory-augmented chat assistant systems can be viewed as instantiations of this framework. An alternative mathematical formulation is presented in Appendix C. For its conciseness, the rest of this paper follows this section's formulation.

### 4.2 LONG-TERM MEMORY SYSTEM: DESIGN CHOICES

We identify four crucial control points for long-term memory of chat assistants, as illustrated in Figure 4. We analyze design choices from existing works and their limitations, and propose our optimizations. Due to space constraints, we present these designs at a high level here, with detailed designs further described in §5 and Appendix D.

**CP 1: Value**   The value represents the format and granularity of each session stored in memory. Given that user-AI chat sessions are often lengthy and cover multiple topics, storing each session as a single item can hinder effective retrieval and reading. Conversely, compressing sessions into summaries or user-specific facts, as seen in prior work such as Zhong et al. (2024) and Du et al. (2024), can lead to information loss, harming the performance of the system to answer detailed questions. In §5.2, we compare three value representation strategies: storing entire sessions, decomposing sessions into individual rounds, and further applying summary/fact extraction.

**CP 2: Key**   Even when sessions are decomposed and compressed, each item still contains substantial information, with only a fraction relevant to the user's query. Therefore, using the value itself as the key, a common practice in prior works (Zhong et al., 2024; Maharana et al., 2024), may be suboptimal. We introduce a key expansion approach in §5.3, where summaries, keyphrases, user facts, and timestamped events are extracted from the values to augment the index. This optimization highlights the key information and enables effective retrieval with multiple pathways.

**CP 3: Query**   For straightforward user queries, the aforementioned key-value optimizations may yield high retrieval accuracy. However, when queries involve temporal references (e.g., "Which restaurant did you recommend last weekend?"), naive similarity search proves insufficient. We address this with a time-aware indexing and query expansion strategy, where values are indexed with timestamped events, and retrieval is restricted to items within the relevant time range (§5.4).

Table 2: A comparison of nine memory-augmented frameworks through the lens of the proposed unified framework. We provide detailed references and discussions of each work in Appendix C. For ChatGPT and Coze, we skip several designs as they are unknown.

| Method | Value | Key | Query | Retrieval | Time-aware | Reading |
|---|---|---|---|---|---|---|
| In-context RAG (Shi et al., 2024) | round/session | K = V | question | flat | No | direct |
| MemoryBank (Zhong et al., 2024) | summary + round | K = V | question | flat | Yes | direct |
| LD-Agent (Li et al., 2024) | summary + fact | K = V | keyphrase | flat | Yes | direct |
| CoN (Yu et al., 2023) | round/session | K = V | question | flat | No | CoN |
| ChatGPT | fact | - | - | - | No | - |
| Coze | fact | - | - | - | No | - |
| RAPTOR (Sarthi et al., 2024) | round/session | node summary | question | flat/interactive | No | - |
| MemWalker (Chen et al., 2023a) | round/session | node summary | question | interactive | No | interactive |
| HippoRAG (Gutiérrez et al., 2024) | round/session | entity | entity | PPR | No | direct |
| Our Design | round | K = V + fact | question + time | flat | Yes | CoN |

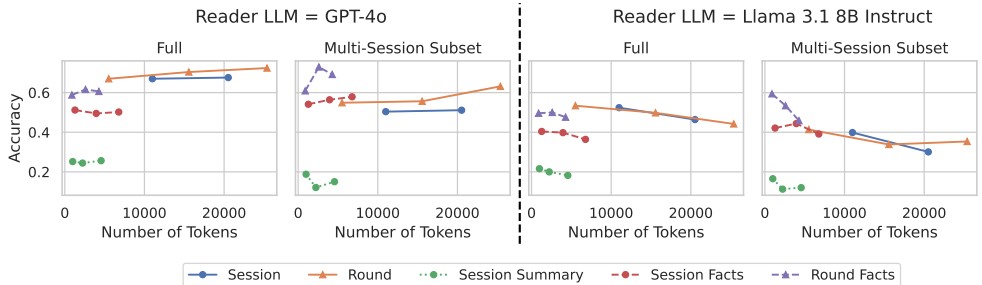

Figure 5: QA performance on LONGMEMEVAL$_M$ with different value designs. Decomposing sessions into rounds improves the QA performance. For the multi-session reasoning questions, further representing the values with the extracted facts improves the QA accuracy.

**CP 4: Reading Strategy**  Answering complex queries may require recalling numerous memory items. Although the retrieval accuracy can be enhanced through the preceding designs, it does not guarantee that the LLM can effectively reason over the extensive context (Shi et al., 2023; Liu et al., 2024). In §5.5, we explore reading strategies and demonstrate that optimizations such as extracting key information before answering (Chain-of-Note, (Yu et al., 2023)) and using structured format prompting (Yin et al., 2023) are crucial for achieving high reading performance.

# 5 EXPERIMENT RESULTS

## 5.1 EXPERIMENTAL SETUP

We mainly study three LLMs: GPT-4o, Llama 3.1 70B Instruct, and Llama 3.1 8B Instruct[3]. For the retriever, we choose dense retrieval with the 1.5B Stella V5 model (Zhang, 2023), given its high performance on MTEB (Muennighoff et al., 2023). Extensive comparisons between Stella V5 and alternative retrievers are provided in Appendix E.2. For the indexing stage, we employ Llama 3.1 8B Instruct to extract summaries, keyphrases, user facts, and timestamped events. When sessions or rounds are used as the key, we only keep the user-side utterances. In the reading stage, the retrieved items are always sorted by their timestamp to help the reader model maintain temporal consistency. Throughout §5.2 to §5.4, we apply Chain-of-Note and json format (discussed in §5.5) by default.

## 5.2 VALUE: DECOMPOSITION IMPROVES RAG PERFORMANCE

Using LONGMEMEVAL$_M$, we compare different value choices in a budget-aware manner. As shown in Figure 5, decomposing sessions into rounds significantly enhances reading performance with GPT-4o as the reader, while performing similarly to non-decomposed sessions when using Llama 3.1 8B Instruct as the reader. However, despite their efficiency in token usage, replacing sessions or rounds with extracted summaries or facts negatively impacts QA performance due to information loss. The only exception is with multi-session reasoning questions, where fact decomposition consistently improves performance. We hypothesize this is because fact decomposition extracts

---

[3]More results on five additional LLMs are reported in Appendix E.1.

Table 3: Retrieval and end-to-end QA performance on LONGMEMEVAL$_M$ with different key designs for indexing. L3.1 = Llama 3.1 Instruct. We find applying document expansion with the extracted user facts (row K = V + fact) greatly improves both retrieval and the downstream QA.

| Key Design | Retrieval | | | | End-to-End QA | | | | | |
| | Metrics@5 | | Metrics@10 | | GPT-4o | | L3.1 70B | | L3.1 8B | |
| | Recall | NDCG | Recall | NDCG | Top-5 | Top-10 | Top-5 | Top-10 | Top-5 | Top-10 |
| --- | --- | --- | --- | --- | --- | --- | --- | --- | --- | --- |
| Value = Round | | | | | | | | | | |
| K = V | 0.582 | 0.481 | 0.692 | 0.512 | 0.615 | 0.670 | 0.600 | 0.624 | 0.518 | 0.534 |
| K = fact | 0.530 | 0.411 | 0.654 | 0.449 | 0.588 | 0.664 | 0.564 | 0.610 | 0.510 | 0.534 |
| K = keyphrase | 0.282 | 0.159 | 0.392 | 0.303 | 0.425 | 0.489 | 0.404 | 0.450 | 0.378 | 0.432 |
| K = V + fact | **0.644** | **0.498** | **0.784** | **0.536** | **0.657** | **0.720** | **0.638** | **0.682** | **0.566** | **0.572** |
| K = V + keyphrase | 0.478 | 0.359 | 0.636 | 0.410 | 0.541 | 0.652 | 0.538 | 0.620 | 0.472 | 0.524 |
| Value = Session | | | | | | | | | | |
| K = V | 0.706 | 0.617 | 0.783 | 0.638 | 0.670 | 0.676 | 0.592 | 0.570 | 0.524 | 0.464 |
| K = summary | 0.572 | 0.448 | 0.648 | 0.468 | 0.554 | 0.252 | 0.498 | 0.512 | 0.444 | 0.216 |
| K = fact | 0.642 | 0.524 | 0.814 | 0.571 | 0.644 | 0.512 | 0.544 | 0.550 | 0.470 | 0.404 |
| K = keyphrase | 0.482 | 0.375 | 0.576 | 0.401 | 0.618 | 0.498 | 0.440 | 0.450 | 0.388 | 0.414 |
| K = V + summary | 0.689 | 0.608 | 0.749 | 0.624 | 0.658 | 0.666 | 0.568 | 0.560 | 0.518 | 0.494 |
| K = V + fact | **0.732** | **0.620** | **0.862** | **0.652** | **0.714** | **0.700** | 0.588 | **0.584** | **0.530** | 0.490 |
| K = V + keyphrase | 0.710 | 0.587 | 0.768 | 0.602 | 0.665 | 0.672 | **0.590** | 0.566 | 0.526 | **0.508** |

the same type of information across all sessions in a more uniform and simplified format, aiding retrieval and reading (Chen et al., 2023b). Finally, we observe that the optimal token budget varies by the reader's capability: while Llama 3.1 8B Instruct's performance drops sharply beyond 3k retrieved tokens, GPT-4o continues to improve even with over 20k retrieved tokens.

## 5.3 KEY: MULTI-KEY INDEXING IMPROVES RETRIEVAL AND RAG

In Table 3, we explore whether summaries, keyphrases, or user facts condensed from the value can serve as better keys than the value itself. Interestingly, despite their more focused semantics, using these condensed forms alone does not enhance the memory recall performance. We hypothesize that this is due to the retriever's ability to already effectively handle long-text semantics.

To leverage both the highlighted information from compression and the completeness of the original value, we applied a simple document expansion technique (Tao et al., 2006; Efron et al., 2012), where the compressed information is concatenated with the original value to form the key during indexing[4]. This approach, particularly when using user facts, yielded an average improvement of 9.4% in recall@$k$ and 5.4% in final accuracy across all models. In Appendix E.2, we further analyze different retrievers and find with alternative retrievers, key expansion with summary and keyphrases could improve Recall@5 when session is used as the value granularity. These results suggest that multi-pathway retrieval can significantly enhance memory recall performance. In the following section, we will investigate the time constraint as another pathway to leverage.

## 5.4 QUERY: TIME-AWARE QUERY EXPANSION IMPROVES TEMPORAL REASONING

A key challenge highlighted by LONGMEMEVAL in building real-world assistant systems is the need to utilize temporal information present in both metadata and user utterances to correctly answer time-sensitive queries. To address this need, we introduce a simple yet effective *time-aware indexing and query expansion* scheme. Specifically, values are additionally indexed by the dates of the events they contain. During retrieval, an LLM $\mathcal{M}_T$ extracts a time range for time-sensitive queries, which is used to filter out a large number of irrelevant values.

As shown in Table 4, this simple design improves recall by an average of 11.3% when using rounds as the value and by 6.8% when using sessions as the value. This improvement remains consistent when key expansion is applied during indexing. We also find that the effectiveness of this method depends on using a strong LLM for $\mathcal{M}_T$ to accurately infer time ranges from queries. Llama 8B, on

---

[4]We also experimented with merging at the retrieval stage by combining the ranks from different pathways, but it underperformed compared to indexing-stage merging. See Appendix E.3 for details.

Table 4: Retrieval performance on the temporal reasoning subset of LONGMEMEVAL$_\text{M}$. Time-aware query expansion significantly facilitates retrieval by narrowing down the retrieval scope.

| Key Setting | Value = Session | | | | Value = Round | | | |
|---|---|---|---|---|---|---|---|---|
| | Metric@5 | | Metric@10 | | Metric@5 | | Metric@10 | |
| | Recall | NDCG | Recall | NDCG | Recall | NDCG | Recall | NDCG |
| (1) K = V | 0.639 | 0.630 | 0.651 | 0.707 | 0.421 | 0.462 | 0.499 | 0.511 |
| (1) w/ Query Expansion ($\mathcal{M}_T$ = GPT-4o) | 0.654 | 0.660 | 0.707 | 0.679 | 0.451 | **0.565** | 0.495 | 0.538 |
| (1) w/ Query Expansion ($\mathcal{M}_T$ = Llama 3.1 8B Instruct) | 0.624 | 0.627 | 0.647 | 0.692 | 0.384 | 0.448 | 0.489 | 0.488 |
| (2) K = V + fact | 0.684 | 0.688 | 0.721 | **0.782** | 0.489 | 0.500 | 0.550 | 0.598 |
| (2) w/ Query Expansion ($\mathcal{M}_T$ = GPT-4o) | **0.722** | **0.732** | **0.797** | 0.758 | **0.526** | 0.548 | **0.722** | **0.669** |
| (2) w/ Query Expansion ($\mathcal{M}_T$ = Llama 3.1 8B Instruct) | 0.677 | 0.688 | 0.711 | 0.744 | 0.481 | 0.532 | 0.570 | 0.647 |

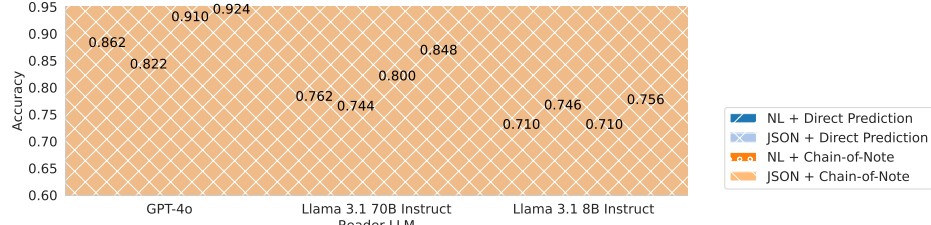

Figure 6: Question answering performance under the oracle retrieval setting. CoN with JSON format outperforms the other three parameter combinations by a large margin. NL = Natural Language.

the other hand, struggles to generate accurate time ranges, often hallucinating or missing temporal cues even with numerous in-context examples. Further analysis is provided in Appendix E.4.

## 5.5 IMPROVING READING WITH CHAIN-OF-NOTE AND STRUCTURED FORMAT

As LONGMEMEVAL requires syntheses across multiple sessions, even an optimal memory retrieval mechanism needs to return a long context to capture all relevant information. To enhance the model's ability to handle long retrieved contexts, we apply two key optimizations. First, we present retrieved items in a structured JSON format (Yin et al., 2023), which helps the model clearly recognize memory items as the data for reading. Additionally, we apply the Chain-of-Note (CoN) reading approach (Yu et al., 2023), instructing the LLM to first extract information from each memory item and then reason based on these notes. This effectively decomposes long-context reading into two simpler subtasks: copying important details and reasoning with more concise notes.

In Figure 6, we evaluate the reading designs under the oracle retrieval setting, where only evidence sessions are provided. Surprisingly, even with perfect retrieval, a suboptimal reading strategy results in up to a 10-point absolute performance drop compared to the best approach for GPT-4o. Notably, when CoN is not applied, JSON format does not consistently outperform the natural language format. However, with CoN, JSON format consistently benefits reader LLMs of various capabilities. Appendix E.5 further analyzes error patterns of LLMs with the enhanced reading strategy.

## 6 CONCLUSION

In this paper, we introduced LONGMEMEVAL, a comprehensive and challenging benchmark designed to evaluate the long-term memory abilities of chat assistants across five core memory tasks: information extraction, multi-session reasoning, temporal reasoning, knowledge updates, and abstention. Through extensive experiments with both commercial systems and long-context LLMs, we demonstrated the significant challenges posed by LONGMEMEVAL, with current systems exhibiting substantial performance drops. By analyzing key design choices across indexing, retrieval, and reading stages, we proposed effective strategies such as session decomposition, fact-augmented key expansion, and time-aware query expansion, which collectively improved both memory recall and the question answering performance. Our findings highlight the need for more sophisticated memory mechanisms to achieve personalized and reliable conversational AI, and LONGMEMEVAL offers a valuable benchmark to drive future advancements in long-term memory capabilities for chat assistants.

## REPRODUCIBILITY STATEMENT

In the paper writing and the subsequent code release, we are committed to enabling other researchers to easily leverage our resources, replicate our results, and build upon our findings. We have documented the benchmark construction process in detail in §3.2 and Appendix A.1, including all the attributes and instructions used to prompt LLMs. We have created and will publicly release the two fixed evaluation datasets, LONGMEMEVAL$_S$ and LONGMEMEVAL$_M$. In addition, we will also release the algorithm and source mixture used to create these two datasets, so that future studies could build upon them to create chat histories of any length. Finally, we have meticulously documented all the implementation details of our memory optimization in Appendix D, and our code will be released along with the benchmark as well. We believe that these efforts for transparency can help advance the field and foster future research endeavors.

## ETHICS STATEMENT

The major artifact released by this work is the LONGMEMEVAL evaluation dataset. To construct the chat history, we utilize ShareGPT[5], which has an Apache 2.0 license, and UltraChat[6], which has an MIT license. In the data release, we plan to use an MIT license. Since the questions and the evidence sessions are newly created, we conducted a rigorous human screen process to ensure that the new dataset does not contain personally identifiable information or offensive content. This paper involves human annotators in two places: (1) the dataset construction and filtering (§3.2) and (2) the manual analysis of commercial systems (§3.4). The process was mainly conducted by three expert annotators, who are in-house NLP researchers that have more than three years of NLP research experience. All the annotators are properly briefed with the annotation objective, and a discussion among them is conducted to resolve uncertain cases. In total, approximately 400 human hours are spent on the dataset construction and 150 hours on the study of commercial systems. The annotators are paid biweekly or monthly, and their salaries include the working hours dedicated to annotation. The data collection protocol has been determined exempt by the IRB of the author's institution. Finally, LONGMEMEVAL and the studied memory system could induce several societal impacts. Storing and recalling user information could cause personal information leakage, and the lack of a memory "deletion" operator could harm the system's trustworthiness. It is also possible that the memory mechanism could be leveraged by malicious parties to toxic contents into the datastore, which may cause a jailbreak behavior during inference. To mitigate such behaviors, it is essential to implement moderation mechanisms that monitor the read/write data flow of the memory. We encourage producers of memory-augmented assistant systems to be aware of these potential harmful effects and apply thorough testing and mitigation.

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

## A  Supplemental Details for LongMemEval

### A.1  Dataset Construction

In this section, we discuss the details of our benchmark construction process.

**Attribute Ontology**  In Table 5, we provide the full attribute ontology used in LongMemEval. This attribute ontology is constructed manually to reflect commonly mentioned topics in user-assistant chats. Five major categories are included: demographic information, lifestyle, situational context, life events, and belongings.

Table 5: Human-constructed user attribute ontology. Each attribute represents a unique dimension of human experience along which a user biography could be constructed. For LongMemEval, we sample user backgrounds based on each of the attributes and construct questions on top of them.

---

**Attribute 1: Demographic Information**
age, gender, ethnicity, nationality, language, education level, occupation

**Attribute 2: Lifestyle**
**2.1: Shopping**: online shopping frequency, favorite stores, loyalty program, sales events, coupons, gift purchasing habits, eco-friendly product preferences, luxury vs budget shopping, technology gadget purchasing, fashion and apparel, grocery shopping, shopping for others
**2.2: Media Consumption**: book, movie, tv show, music, podcast, video game, streaming service, theater, magazine and newspaper, youtube, educational content, audiobook and e-book
**2.3: Social Media Engagement**: posting, commenting, followers, groups, hashtags, campaigns, messaging, live streaming, social media breaks
**2.4: Daily Routines**: wake-up time, bedtime, work or school start time, meal time, exercise routines, coffee or tea break, commuting, evening activities, weekend routines, cleaning schedules, time spent with family or friends
**2.5: Travel**: frequency, destination, road trips, travel agencies, outdoor adventures, airlines, hotel, travel with family vs solo travel, packing habits
**2.6: Recreation**: reading, painting, musical instruments, dancing, watching sports, participating in sports, gardening, bird watching, fishing or hunting, board games, video games, fitness classes, yoga, sculpting, photography, stand-up comedy, writing, collecting, model building, aquarium keeping
**2.7: Eating and Cooking**: home cooking, food delivery, vegetarian or vegan, favorite cuisines, snacking habits, barbecue, baking, cocktail mixing, cooking classes
**2.8: Event Participation**: concerts, theater, galleries and museums, sports games, film festivals, religious services, book readings, charity events, trade shows, lectures or workshops, theme parks, local markets, networking events, sports, auto racing, workshops, museum tours

**Attribute 3: Situational Context**
**3.1: Home**: living room, kitchen, bathroom, room style, room lighting, furniture, technology, plants
**3.2: Social Context**: alone, family, friends, interactions with strangers
**3.3: Time Context**: time of day, day of week, seasonal

**Attribute 4: Life Events**
graduations, academic achievements, study abroad, significant academic projects, job promotions, starting a business, births and adoptions, marriages, family reunions, illness or surgeries, mental health journeys, purchasing a home, trips, movement, living abroad, refugee or immigration, loss of loved ones, name change, belief, milestone

**Attribute 5: Belongings**
cars, bikes, vehicles, computer, phone, pet, farm animal, animal care items, home, land, art, antiques, collectible, rare items, clothing, jewelry, shoes, bag, sports gear, musical instruments, health related devices, crafting, photography

---

**Background Sampling**  Based on each of the attributes, we prompt Llama 3 70B Instruct to generate a background paragraph outlining the user memory and experience. In our preliminary studies, we find that the following zero-shot prompt in Figure 7 can already guide the model to generate a long and focused user background with sufficient details, which suffices for the next step of question creation. We thus use the same prompt for the final version of LongMemEval.

I will give you a topic. Please imagine you are a user that wants to recall and record recent personal facts along the topic. Generate a long text describing these personal facts. Use your imagination and generate the personal facts. Make it long and involve several recent facts or recent events spanning many days, weeks, or monthes. You may state the facts in plain language and no need to make it story-like.

Topic: {attribute}

Recent Personal Facts related to {attribute}:

Figure 7: The prompt for constructing user backgrounds based on an attribute.

I will give you a past memory. Use the memory to act as a normal user to chat with a chat assistant. In the chat, you may ask it to assist you various tasks or ask it about various information. However, make sure that your convey the following fact about you: "{evidence_statement}". In addition, make sure your message is concise (1-2 simple sentences), since the real users often do not bother write a long message. I will provide you with the chat history and the response from the assistant. Directly generate the next response from the user's perspective. You must simulate the tone of a neutral user and do not be overly enthusiastic, verbose, formal, or polite. For conciseness, DO NOT react to the assistant's message with e.g., "thanks" or "I will do that". Instead, directly state the follow-up questions or new questions.

Memory: {background}

Chat History:

assistant: Hi! How can I assist you today?

... (more rounds as the conversation continues) ...

Figure 8: The prompt for instruction an LLM to act as a user and initiates a task-oriented dialogue with another LLM. Both the background and the evidence statement is provided. This prompt is used for the question type **single-session**. For the other question types, the prompt components are slightly different but the prompt overall follows the same style.

**Question Construction**  As discussed in §3.2, based on the generated user backgrounds, Llama 3 70B Instruct is used to propose question and answers for each question type. Additionally, for the question type **temporal-reasoning**, **multi-session reasoning**, and **single-session-preference**, we use GPT-4o to propose several questions. Nevertheless, we find most of the questions to be unsatisfactory and manually filter and edit most of the questions. In total, approximately 1000 questions were generated for each question type, and the final yield rate is about 5%. For each question, we then manually decompose the answer into the evidence statements. If the question or the evidence statements involve time mentions, we assign a timestamp to the question and the evidence statements at this stage. Note that if timestamps are specified for the evidence statements at this stage, these timestamps will always be used for corresponding evidence sessions. Otherwise, the timestamps will be randomly assigned at the history construction stage with all the other sessions.

**Evidence Session Construction**  Using the question and the decomposed evidence statements, we use Llama 3 70B Instruct to simulate one user-AI chat history per evidence statement via self-chat. In Figure 8, we present an example of the chat simulation prompt, where we ask the user LLM to indirectly mention the evidence statement while avoiding to talk about other evidence statements for the question, if there are any. We include two crucial instructions in the prompt to make sure (1) the evidence statement is provided in an indirect way and (2) the generated messages are concise and

thus mimic the style of user messages. On the assistant side, we directly provide the input generated by the user LLM without any prompt engineering. We simulate the chat for 10 round at maximum and stopped prematurely when either side of the LLMs generates an empty sequence indicating the end of the conversation.

Subsequently, expert annotators manually inspect and edit each of the generated sessions to ensure that (1) the required evidence statements are present in the conversation, (2) no other evidence statements are leaked into the conversation, (3) the evidence statements are provided in a colloquial style, especially for the data and time mentions, and (4) the conversation ends gracefully. In total, roughly 70% of the sessions are human edited. We note that in a few rare instances, the user LLM fails by assuming the assistant role instead. When these failures are identified, we discard the instance if the conversation cannot be fixed.

## A.2 HISTORY CONSTRUCTION

In order to construct a coherent and freely extensible chat history, we design a three-staged pipeline that include *session pool construction*, *session sampling*, and *timestamp resolution*.

**Session pool construction**  For each question, we draw the history sessions from three sources: ShareGPT (Zheng et al., 2023), UltraChat (Ding et al., 2023), and the simulated sessions corresponding to other attributes using the same pipeline mentioned in the previous section. This pool ensures that the non-evidence history sessions have similar topic or format as the evidence sessions, while avoiding providing conflicting information that would invalidate the question.

**Session sampling**  To sample a history containing $x$ sessions, we randomly sample from the aforementioned three sources and shuffle the sessions together with the question's evidence sessions. For LONGMEMEVAL, we always use the following mixture: 25% ShareGPT, 25% UltraChat, and 50% simulated sessions. If the evidence sessions need to follow a specific order, we swap their orders accordingly after shuffling.

**Timestamp resolution**  Finally, we randomly assign timestamps to the session following their order of the history. If the evidence sessions are associated with pre-defined timestamps, we use them as anchors to determine the range of timestamp of the non-evidence sessions preceding or following them. Other wise, we randomly assign tiemstamps in May 2023.

## A.3 BASIC STATISTICS

In Figure 9, we present the basic statistics of LONGMEMEVAL, revealing that most questions require evidence from multiple sessions (up to six) and that evidence statements are positioned diversely within sessions, increasing the challenge to the memory design.

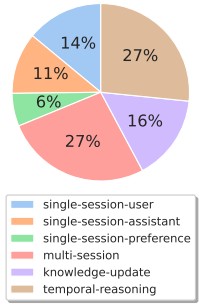

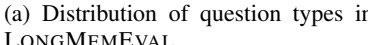

(a) Distribution of question types in LONGMEMEVAL.

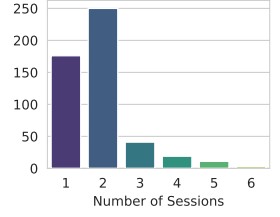

(b) Distribution of the number of evidence sessions. Most questions emphasize multi-session reasoning, requiring reading up to six sessions to answer.

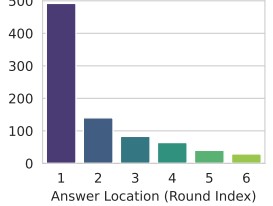

(c) Distribution of the location of the evidence statement within the evidence sessions. Most evidence statements are located at the beginning of the chat.

Figure 9: LONGMEMEVAL challenges chat assistants through its (a) diverse question types, (b) emphases on multi-session reasoning, and (c) diverse evidence locations within sessions.

## A.4 EVALUATION METRIC BUILDING

To accurately evaluate the diverse responses of LLMs, we use an expert-written prompt to instruct GPT-4o as the correctness judge. We present the full prompt in Figure 10. To enable the model to handle detailed edge cases as how expert evaluators would do, we design separate prompts for a number of tasks. To ensure the prompt has a high agreement with expert judge, we sample 30 questions per problem type, collect the long-context generation results from GPT-4o and Llama 3.1 8B Instruct, and report the judgment correctness by category.

As shown in Table 6, the prompt-engineered GPT-4o judge achieves reliable performance in evaluating both GPT-4o and Llama-3.1-8B-Instruct as the generation model. The evaluator's judgment slightly deviates from human experts for the single-session-preference and abstention problems due to the open-ended nature of the response. Nevertheless, our evaluation prompt still achieves 90% or higher accuracy under all settings. We will include this prompt in the benchmark package that we will release to enable consistent comparisons for future work.

---

**temp-reasoning**

I will give you a question, a correct answer, and a response from a model. Please answer yes if the response contains the correct answer. Otherwise, answer no. If the response is equivalent to the correct answer or contains all the intermediate steps to get the correct answer, you should also answer yes. If the response only contains a subset of the information required by the answer, answer no. In addition, do not penalize off-by-one errors for the number of days. If the question asks for the number of days/weeks/months, etc., and the model makes off-by-one errors (e.g., predicting 19 days when the answer is 18), the model's response is still correct.

---

**knowledge-update**

I will give you a question, a correct answer, and a response from a model. Please answer yes if the response contains the correct answer. Otherwise, answer no. If the response contains some previous information along with an updated answer, the response should be considered as correct as long as the updated answer is the required answer.

---

**single-session-preference**

I will give you a question, a rubric for desired personalized response, and a response from a model. Please answer yes if the response satisfies the desired response. Otherwise, answer no. The model does not need to reflect all the points in the rubric. The response is correct as long as it recalls and utilizes the user's personal information correctly.

---

**Other question types**

I will give you a question, a correct answer, and a response from a model. Please answer yes if the response contains the correct answer. Otherwise, answer no. If the response is equivalent to the correct answer or contains all the intermediate steps to get the correct answer, you should also answer yes. If the response only contains a subset of the information required by the answer, answer no.

Figure 10: Evaluation instructions for the GPT-4o judge. We provide the question, answer, and the model's hypothesis after the instruction and ask GPT-4o to directly generate "yes" or "no".

Table 6: Meta-evaluation results of prompt-engineered GPT-4o judge. We observe a high evaluation accuracy across all the problem types in LONGMEMEVAL.

| Question Type | Answer Model | |
|---|---|---|
| | GPT-4o | Llama-3.1-8B-instruct |
| single-session-user | 1.00 (30/30) | 0.97 (29/30) |
| single-session-assistant | 1.00 (30/30) | 1.00 (30/30) |
| single-session-preference | 0.90 (27/30) | 0.97 (29/30) |
| multi-session | 1.00 (30/30) | 1.00 (30/30) |
| knowledge-update | 1.00 (30/30) | 1.00 (30/30) |
| temporal-reasoning | 1.00 (30/30) | 0.97 (29/30) |
| abstention | 0.97 (29/30) | 0.90 (27/30) |
| Average | 0.98 | 0.97 |

# B    A HUMAN STUDY ON COMMERCIAL MEMORY CHATBOTS

We evaluate two commercial memory-augmented chatbots: ChatGPT (OpenAI, 2024) and Coze (Coze, 2024). We randomly selected 97 questions and created a short chat history of 3-6 sessions by sampling according to a mixture of 50% ShareGPT and 50% simulated sessions. We skip two type of questions that the assistants cannot answer: (1) a subset of temporal-reasoning, since our manual analysis cannot afford interacting with the (potentially evolving) systems across multiple months, and (2) single-session-assistant, since the systems do not remember any information given by the assistant. We also did not evaluate the abstention ability of these two systems because an early version of the dataset without any abstention questions was used for this analysis.

Since these systems only support memory features via their web interfaces, human annotators manually interacted with the chat assistants session-by-session, ending with a new session where the question was posed. After collecting the model's response, the annotator manually evaluates the answer's correctness. Finally, to start evaluating the next instance from a fresh state, the annotator manually clears the model's memory through the web interface. We distribute the questions across five annotators. A discussion is performed among the annotators whenever there is a concern with the model's response or with the evidence sessions. All evaluations were conducted in the first two weeks of August 2024.

In Table 7, we present the detailed human evaluation results by problem types. We observe that when the task is memorizing the information from a single session (column IE), both systems can answer a considerable number of problems correctly. However, for the other question types where aggregation across multiple sessions is generally required, both systems exhibit significant performance drops. Compared to ChatGPT, we find most of Coze's errors are due to failing to record information from some session. On the other hand, ChatGPT generally records the evidence statements immediately after it has been presented in the evidence session. However, as the interaction proceeds, ChatGPT often modify this information when it compresses the history, resulting in information loss. This highlights the potential trade-off between reliable personalization and efficiency.

Table 7: Human evaluation results of two systems categorized by evaluated ability types. We use the questions from single-session-user for the IE column. For the temporal reasoning column (TR), we use the questions that do not require reasoning with metadata.

| System | Model | Memory Ability | | | |
|---|---|---|---|---|---|
| | | IE | MR | KU | TR |
| ChatGPT | GPT-4o-mini | 1.000 | 0.647 | 0.667 | 0.652 |
| | GPT-4o | 0.688 | 0.441 | 0.833 | 0.435 |
| Coze | GPT-3.5-turbo | 0.625 | 0.118 | 0.375 | 0.043 |
| | GPT-4o | 0.813 | 0.147 | 0.208 | 0.391 |

# C UNIFIED MEMORY VIEW

## C.1 AN ALTERNATIVE MATHEMATICAL FORMULATION

In this section, we provide a more rigorous mathematical formulation of the unified memory framework for interested readers. Since we find the corresponding natural language descriptions clear enough for presenting the main arguments while avoiding introducing excessive symbols.

We formulate long-term memory as a massive key-value datastore $D = [(k_1, v_1), (k_2, v_2), ...]$, where the keys $k_i$ can be heterogeneous and the values $v_i$ might repeat. More concretely, a key could be discrete or continuous. In the discrete case, the key could be a sentence, a paragraph, a fact, or an entity, etc. In the continuous case, the key could be e.g., the model's internal representation under some inputs. For the three stages, we formulate them with four functions: $\mathcal{I}, \mathcal{Q}, \mathcal{S}, \mathcal{R}$. The offline index function $\mathcal{I}(S, D)$ converts a session $S$ into a series of key-value tuples. Alternatively, the online index function $\mathcal{I}_{online}(S, D)$ updates D with the key-value tuples extracted from S, potentially removing or editing the existing items in D. The query formulation function $\mathcal{Q}(q)$ converts a natural language user query $q$ into a representation $q'$ that function $\mathcal{S}$ could leverage. The salience scoring function $\mathcal{S}(q', D)$ orders D by their relevance to $q'$. Potential initializations of $\mathcal{S}$ include ranking dense text embedding, graph-based ranking, and so on. Finally, the reading function $\mathcal{R}(M, D')$ combines the model M with D', the most relevant part as ranked by the salience scoring function $\mathcal{S}$ and generates a response. The methods outlined in the columns of Table 2 could be seen as different ways to instantiate the four functions $\mathcal{I}, \mathcal{Q}, \mathcal{S}, \mathcal{R}$.

## C.2 EXISTING MEMORY SYSTEMS FROM THE UNIFIED VIEW

In §4, we have presented a unified view of the memory systems with three stages and four control points. In this section, we provide a closer look at nine memory-augmented chat systems and view them as specific instances of the unified framework. Specifically, we consider in-context RAG (Shi et al., 2024), Memorybank (Zhong et al., 2024), LD-Agent (Li et al., 2024), CoN (Yu et al., 2023), ChatGPT, Coze, RAPTOR (Sarthi et al., 2024), MemWalker (Chen et al., 2023a), and HippoRAG (Gutiérrez et al., 2024). Table 2 provides an overall comparison between the systems, among which we also situate the recommended design identified through the paper's empirical study.

**Indexing** For the value representation, most of the surveyed systems either directly use the sessions themselves or use a mixture of them with the extracted facts or summaries. Only three systems (LD-Agent, ChatGPT, and Coze) use only the compressed values to replace the original ones, which may incur an information loss. When it comes to the indexing key, we observe three major types of decisions. First, a number of systems (in-context RAG, Memorybank, LD-Agent, and CoN) simply use the values as the keys, which wins in terms of simplicity. By comparsion, HippoRAG builds an entity-centric index, and RAPTOR/Memwalker build a hierarchical index using recursive summarization. It is noteworthy that while a more complex memory indexing structure can potentially benefit certain types of queries, they also increase the cost of creating and maintaining the index in online interactions. Specifically, for HippoRAG, RAPTOR, and Memwalker, some level of re-indexing is required when a new session is added to the memory, increasing the computational overhead of these sytems.

**Retrieval** For most of the systems, we find that the question is generally used as the query, which is intuitive since the questions are generally short. LD-Agent and HippoRAG extract additional information and leverage them to better pinpoint entity-centric knowledge within the index. In terms of the query-key matching process, most systems use a flat retrieval setup, where a similarity search is directly conducted between the query and the keys. HippoRAG uses Personalized PageRank (PPR) to leverage its entity graph structure to match passages that feature entities close to the seed entities extracted from the query. Different from these approaches, MemWalker bridges the retrieval and inference through an interactive reading mechanism where the reader LLM self-navigates the indexing structure to find the answer. This method brings the additional robustness in case the retriever fails, but also incurs additional latency costs due to more LLM inference.

**Generation** Most of the systems apply a direct reading mechanism where the reader model is provided with the retrieved memory items and directly asked to generate the response. However,

as we show in the main results, adapting an extract-before-read strategy is important to a high performance. On the other hand, the interactive reading mechanism such as that used in MemWalker allows the reader to backtrack and search again in case the recalled memory is inadequate.

# D    MEMORY OPTIMIZATIONS: IMPLEMENTATION DETAILS

In this section, we describe our implementations of the memory optimizations in detail.

**Value Decomposition**    We design an LLM-based method for compressing the value, which are either sessions or rounds, into three different formats: summaries, keyphrases, or user facts. In Figure 11, we present the corresponding zero-shot and few-shot prompts. Note that few-shot learning is only applied for user fact extraction, and we find it to be unnecessary for the rest two tasks. For in-context examples, we randomly sample ten examples, five from the simulated user sessions and five from ShareGPT. The expected responses are generated by GPT-4o and modified by human annotators. We manually tune the prompts by observing several examples. Llama 3.1 8B Instruct is used for all the extraction experiments. Note that for all the experiments, we only provide the user-side messages for the extraction, skipping the assistant's responses.

**Key Expansion**    To expand the value documents with summaries, keyphrases, or user facts, we follow the same pipeline in the previous section to first extract these pieces of information from each value. Then, we directly prepend the extracted information to the corresponding key.

**Time-Aware Indexing and Query Expansion**    At the indexing stage, we instruct Llama 3.1 8B Instruct to extract the event mentions in the text whose timestamp could be inferred. At the retrieval stage, we instruct an LLM (we explore GPT-4o and Llama 3.1 8B Instruct) to extract the a time range for retrieval for the problems that specifically focus on a range of time. We present the corresponding prompts in Figure 12. Ten human-written in-context examples are additionally provided.

**Reading Strategy**    We present the prompt for reading in Figure 13, which is a small variation of Chain-of-Note (CoN). Overall, the prompt shares the same idea with CoN, that the model is asked to traverse the documents and extract the evidence before generating the answer to the question. We use greedy search for all the experiments, and set the maximum generation length to 800 tokens.

**Summaries**

Below is a transcript of a conversation between a human user and an AI assistant. Please summarize the following dialogue as concisely as possible in a short paragraph, extracting the main themes and key information. In your summary, focus more on what the user mentioned or asked for. Dialogue content: {session or round}

---

**Keyphrases**

Below is a transcript of a conversation between a human user and an AI assistant. Generate a list of keyphrases for the session. Separate each keyphrase with a semicolon. Dialogue content: {session or round}

---

**User Facts**

You will be given a list of messages from a human user to an AI assistant. Extract all the personal information, life events, experience, and preferences related to the user. Make sure you include all details such as life events, personal experience, preferences, specific numbers, locations, or dates. State each piece of information in a simple sentence. Put these sentences in a json list, each element being a standalone personal fact about the user. Minimize the coreference across the facts, e.g., replace pronouns with actual entities. If there is no specific events, personal information, or preference mentioned, just generate an empty list.

Human user messages: {session}

Personal facts about the user (a list of strings in json format; do not generate anything else):

Figure 11: Zero-shot prompts for extracting information from the value items for indexing. For user fact extraction, we additionally provide ten in-context examples.

**Extracting Timestamped Events**

You will be given a list of messages from a human user to an AI assistant, as well as the time the conversation took place. Extract all events related to the user as long as its date is specified or could be inferred. If the time some event took place cannot be inferred, do not extract that event. Return the events in a json list where each item contains two fields: "date" and "event". Write date in the form YYYY/MM/DD. If there is no specific event, just write an empty list.

---

**Query Expansion**

You will be given a question from a human user asking about some prvious events, as well as the time the question is asked. Infer a potential time range such that the events happening in this range is likely to help to answer the question (a start date and an end date). Write a json dict two fields: "start" and "end". Write date in the form YYYY/MM/DD. If the question does not have any temporal referencea, do not attempt to guess a time range. Instead, just say N/A."

Figure 12: Zero-shot prompt for extracting timestamped events from the value items for indexing and the prompt for extracting time range from the user question. For both settings, we additionally provide ten in-context examples.

**CoN Prompt**

I will give you several history chats between you and a user. Please answer the question based on the relevant chat history. Answer the question step by step: first extract all the relevant information, and then reason over the information to get the answer.

History Chats: {chat_history}

Current Date: {question_date}

Question: {question}
Answer (step by step):

---

**Non-CoN Prompt**

I will give you several history chats between you and a user. Please answer the question based on the relevant chat history.

History Chats: {chat_history}

Current Date: {question_date}

Question: {question}
Answer:

Figure 13: The prompt for reading with recalled memory with and without Chain-of-Note.

# E  EXTENDED ANALYSES

We provide additional analyses on LONGMEMEVAL, including more LLMs, retriever selection, memory optimization, and error analyses of end-to-end retrieval-augmented generation.

## E.1  RESULTS ON MORE LLMs

In Table 8, we evaluate five more models on LONGMEMEVAL: the 1B and 3B models of Llama-3.2 Instruct (Dubey et al., 2024), Qwen-2.5-7B (Team, 2024), Phi-3.5-mini-instruct (Abdin et al., 2024b), and Mistral-Nemo-Instruct-2407 (Mistral AI Team, 2024). Overall, we observe that the performance is consistent with the results reported in the paper. For instance, long-context performance on LONGMEMEVAL$_S$ is significantly lower than the oracle retrieval performance. In addition, the fact-based key expansion and CoN consistently improves the performance.

| Model | Oracle Sessions | | LONGMEMEVAL$_S$ | | | | LONGMEMEVAL$_M$ | |
|---|---|---|---|---|---|---|---|---|
| | LC Direct | LC CoN | LC Direct | LC CoN | K = V | K = V+fact | K = V | K = V+fact |
| Mistral-Nemo-Instruct-2407 | **0.688** | 0.686 | 0.162 | 0.286 | 0.640 | **0.666** | 0.554 | **0.598** |
| Qwen2.5-7B | 0.282 | **0.504** | 0.128 | 0.144 | 0.452 | **0.462** | 0.390 | **0.424** |
| Phi-3.5-mini-instruct | 0.488 | **0.490** | 0.312 | 0.352 | 0.550 | **0.570** | 0.462 | **0.468** |
| Llama-3.2-3B-Instruct | 0.522 | **0.636** | 0.008 | 0.010 | 0.466 | **0.508** | 0.466 | **0.470** |
| Llama-3.2-1B-Instruct | 0.386 | **0.398** | 0.010 | 0.016 | 0.312 | **0.336** | 0.286 | **0.312** |

Table 8: Evaluation results using five additional LLMs of various sizes. "LC" indicates directly reading from the entire history without any memory operations. For the last four columns that have memory operations, we always use round as the value granularity.

## E.2  ABLATIONS ON RETRIEVER SELECTION

In Table 9, we compare the performance of Stella V5 with two alternative popular retrievers: BM25 (Robertson & Zaragoza, 2009) and Contriever (Izacard et al., 2022). We do not report the results of retrievers that use larger embedding models, as their latency is generally unrealistic for real applications. The retrievers are compared under two settings: (1) vanilla key design where the values themselves are used as the key and (2) the key expansion strategies introduced in the main text. We list the key observations below:

- Both dense retrieval embeddings have significantly higher performance than the BM25 sparse retrieval method. The trend is the same across most settings.

- The recommended technique of performing key expansion with user fact consistently improves the performance over directly using the value as the key.

- Key expansion with summary and keyphrases could improve the performance in some settings. For example, both summary and keyphrase improve the Recall@5 of Contriever when session is used as the value granularity. However, expanding the key with facts still gives the greatest performance gain.

## E.3  POST-RETRIEVAL RANK MERGING FOR INDEX EXPANSION

In §5.3, we have introduced and evaluated the key expansion strategy for multi-path retrieval, where important information is extracted from the value items and then used to augment the key. However, an alternative strategy is to directly create a separate key with the retrieved information and place it in parallel as the original keys. At the retrieval stage, the query could be used to retrieve from pools defined multiple types of keys and the values from different sequences could be merged subsequently according to their rank. Effectively, this implements the true "multi-path retrieval. We call this strategy "rank merging". In Table 10, we thoroughly evaluate rank merging verse key merging, the strategy we recommended in the main text. We find that rank merging has much lower performance than key merging. One potential reason is that rank merging increases the index size by $m+1$ times, where $m$ is the number of information pieces extracted from each (key, value) pair. By comparison, key merging highlights the important information while avoiding exploding the size of the index.

Table 9: A comparison between three retrievers under different key-value indexing settings.

| Key Design | Retriever | Value = session | | | | Value = round | | | |
|---|---|---|---|---|---|---|---|---|---|
| | | Metric@5 | | Metric@10 | | Metric@5 | | Metric@10 | |
| | | Recall | NDCG | Recall | NDCG | Recall | NDCG | Recall | NDCG |
| K = V | BM25 | 0.634 | 0.516 | 0.710 | 0.540 | 0.472 | 0.352 | 0.538 | 0.372 |
| | Contriever | 0.723 | 0.634 | 0.823 | 0.663 | 0.589 | 0.454 | 0.747 | 0.495 |
| | Stella V5 1.5B | 0.720 | 0.594 | 0.794 | 0.615 | 0.660 | 0.498 | 0.784 | 0.528 |
| K = V + summary | BM25 | 0.626 | 0.544 | 0.694 | 0.565 | - | - | - | - |
| | Contriever | 0.732 | 0.632 | 0.823 | 0.657 | - | - | - | - |
| | Stella V5 1.5B | 0.689 | 0.608 | 0.749 | 0.624 | - | - | - | - |
| K = V + fact | BM25 | 0.683 | 0.560 | 0.757 | 0.582 | 0.554 | 0.454 | 0.608 | 0.473 |
| | Contriever | 0.762 | 0.632 | 0.862 | 0.658 | 0.612 | 0.489 | 0.756 | 0.530 |
| | Stella V5 1.5B | 0.732 | 0.520 | 0.862 | 0.552 | 0.644 | 0.498 | 0.784 | 0.536 |
| K = V + keyphrases | BM25 | 0.632 | 0.546 | 0.711 | 0.569 | 0.453 | 0.391 | 0.538 | 0.416 |
| | Contriever | 0.740 | 0.638 | 0.849 | 0.668 | 0.546 | 0.450 | 0.672 | 0.489 |
| | Stella V5 1.5B | 0.710 | 0.587 | 0.768 | 0.602 | 0.478 | 0.359 | 0.636 | 0.410 |

Table 10: Retrieval and end-to-end QA performance on LONGMEMEVAL$_M$ with different key designs for indexing. L3.1 = Llama 3.1 Instruct. RM = Rank Merging and KM = Key Merging.

| Key Design | Retrieval | | | | End-to-End QA | | | | | |
|---|---|---|---|---|---|---|---|---|---|---|
| | Metrics@5 | | Metrics@10 | | GPT-4o | | L3.1 70B | | L3.1 8B | |
| | Recall | NDCG | Recall | NDCG | Top-5 | Top-10 | Top-5 | Top-10 | Top-5 | Top-10 |
| Value = Round | | | | | | | | | | |
| K = V | 0.582 | 0.481 | 0.692 | 0.512 | 0.615 | 0.670 | 0.600 | 0.624 | 0.518 | 0.534 |
| K = V + fact (RM) | 0.478 | 0.372 | 0.568 | 0.403 | 0.558 | 0.596 | 0.536 | 0.586 | 0.458 | 0.482 |
| K = V + keyphrase (RM) | 0.386 | 0.376 | 0.536 | 0.329 | 0.485 | 0.590 | 0.460 | 0.558 | 0.426 | 0.466 |
| K = V + fact (KM) | **0.644** | **0.498** | **0.784** | **0.536** | **0.657** | **0.720** | **0.638** | **0.682** | **0.566** | **0.572** |
| K = V + keyphrase (KM) | 0.478 | 0.359 | 0.636 | 0.410 | 0.541 | 0.652 | 0.538 | 0.620 | 0.472 | 0.524 |
| Value = Session | | | | | | | | | | |
| K = V | 0.706 | 0.617 | 0.783 | 0.638 | 0.670 | 0.676 | 0.592 | 0.570 | 0.524 | 0.464 |
| K = V + summary (RM) | 0.608 | 0.488 | 0.684 | 0.511 | 0.600 | 0.620 | 0.510 | 0.544 | 0.468 | 0.466 |
| K = V + fact (RM) | 0.618 | 0.511 | 0.754 | 0.548 | 0.622 | 0.688 | 0.574 | 0.574 | 0.468 | 0.466 |
| K = V + keyphrase (RM) | 0.550 | 0.435 | 0.650 | 0.466 | 0.560 | 0.620 | 0.454 | 0.506 | 0.492 | 0.482 |
| K = V + summary (KM) | 0.689 | 0.608 | 0.749 | 0.624 | 0.658 | 0.666 | 0.568 | 0.560 | 0.518 | 0.494 |
| K = V + fact (KM) | **0.732** | **0.620** | **0.862** | **0.652** | **0.714** | **0.700** | 0.588 | **0.584** | **0.530** | 0.490 |
| K = V + keyphrase (KM) | 0.710 | 0.587 | 0.768 | 0.602 | 0.665 | 0.672 | **0.590** | 0.566 | 0.526 | **0.508** |

## E.4 STRONG AND WEAK LLMS FOR EXTRACTING TIME RANGES FROM QUERIES

In §5.4, we concluded that a strong model is required to accurately extract the time ranges from the questions, even if in-context examples (with balanced labels) are provided. In this section, we further illustrate this finding with examples in Table 11. We find that the Llama 3.1 8B Instruct model can generate a correct temporal range when the time references are clearly defined (example 2). However, in many of the cases where the question does not have any temporal reference (example 1, 2, 4), the model often mistakenly extracts a time range, which erroneously prunes out the search space, leading to a low memory recall. By contrast, GPT-4o is able to refuse to generate a time range when the question does not have a time reference.

## E.5 ERROR ANALYSIS

In this section, we analyze the source of error of various LLMs with our best memory design. Specifically, we use round as the value granularity, expand the key with extracted user facts, and apply CoN as the reading strategy. Stella v5 1.5B is used as the retriever, and top-10 items are provided to the model with a JSON structured prompt. In Figure 14, we analyze the distribution of the correct and error cases for three different reader LLMs. First, we observe that a substantial proportion of errors corresponds to correct retrieval yet wrong generation (15%~19% of all instances, and 40%~50% among the error instances). The proportion is higher when a weaker

Table 11: Example of time span extraction outputs from GPT-4o and Llama 3.1 8B Instruct.

**Example 1**
Question Date: 2023/05/28
Question: How long had I been taking guitar lessons when I bought the new guitar amp?
Predicted time range (GPT-4o): No date extracted **[correct]**
Predicted time range (Llama 3.1 8B Instruct): 2023/05/01~2023/05/28 **[false positive]**

**Example 2**
Question Date: 2023/04/27
Question: Which airline did I fly with the most in March and April?
Predicted time range (GPT-4o): 2023/03/01~2023/04/30 **[correct]**
Predicted time range (Llama 3.1 8B Instruct): 2023/03/01~2023/04/30 **[correct]**

**Example 3**
Question Date: 2023/06/28
Question: How many days before the 'Rack Fest' did I participate in the 'Turbocharged Tuesdays' event?
Predicted time range (GPT-4o): No date extracted **[correct]**
Predicted time range (Llama 3.1 8B Instruct): 2023/06/21~2023/06/28 **[false positive]**

**Example 4**
Question Date: 2023/03/10
Question: Which seeds were started first, the tomatoes or the marigolds?
Predicted time range (GPT-4o): No date extracted **[correct]**
Predicted time range (Llama 3.1 8B Instruct): 2023/03/01~2023/03/10 **[false positive]**

reader LLM is used. This indicates that the reading strategy still has a large room of improvement. In addition, we observe that for the reader LLM to generate a correct answer, performing correct retrieval is necessary in ~90% of the time. We observe that the rest 10% instances are mostly of the question type knowledge-update and the retriever was only able to identify the updated knowledge but failed to retrieved the previous information before update. For these cases, our strict retrieval evaluation criteria will deem that the retrieval has failed. Overall, this result also highlights the high quality of LONGMEMEVAL, as the reader LLM cannot take any shortcut to answer the question correctly without a correct memory recall.

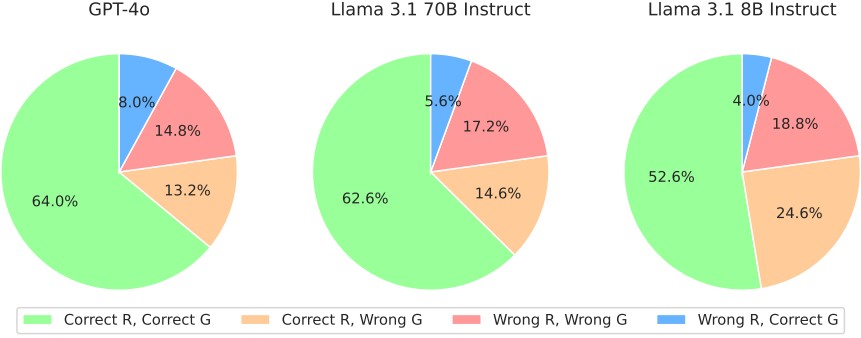

Figure 14: An analysis of the error distribution of different reader models. We use R and G to indicate correct Recall@10 and correct answer based on the top-10 retrieved results.

