# OpenReview forum: "LongMemEval: Benchmarking Chat Assistants on Long-Term Interactive Memory"
_ICLR.cc/2025/Conference — ICLR 2025 Poster_

### Official Review · Reviewer_LhuP · 2024-10-23

**Soundness:** 2
**Presentation:** 1
**Contribution:** 3
**Rating:** 3
**Confidence:** 4

**Summary:**

This paper proposes a 500 LongMemEval dataset focusing on evaluating five crucial long-term capabilities of LLMs: (single-session) information extraction, multi-session reasoning, knowledge update, temporal reasoning, and abstention.
The authors define each conversation as a key-value pair, where the key is the "time index" and the value is the corresponding chat.
They propose a three-stage framework for indexing, retrieval, and reading.
The framework has four control points: key (in the indexing stage), value (in the indexing stage), query (in the retrieval stage), and reading strategy (in the reading stage).
They conduct experiments to test their framework using LongMemEval on GPT-4o and Llama-3 (70B and 8B).

**Strengths:**

1. This paper aims to tackle a problem of LLM from the time perspective, which may be a useful method.
2. Apart from the single-session, the knowledge update and the temporal reasoning are known issues in contemporary LLMs.
3. They study these phenomena in the long-context, long-term scenario.

**Weaknesses:**

This paper has several drawbacks regarding presentation, soundness, and contribution.

TL;DR, the major weaknesses are as follows (see **Details** below for further elaboration):

**Weakness 1.** [Presentation and Soundness] **This paper lacks careful proofreading, as it contains an unusually high number of typos and inconsistent writing. The numerical rounding is also confusing.** For example, in the third contribution, the authors report improvement is 7% ~ 11% in Line 99, but it's 6.7% ~ 11.4% in Lines 480-481 (should report the exact value or *at least* add the term "around" in Line 99). **Additionally, the flow in each section is not strongly connected, requiring significant revision to reach a publishable standard. Many tables and figures need substantial revisions and adjustments. The authors also need to be crisp in their writing.** For one thing, in the main content, the "result and discussion" of their proposed framework only spans two pages (Sections 5.2 to 5.5), including two tables and one figure. Without these, the text is roughly a page only in the main paper, which is insufficient for the ICLR community.

**Weakness 2.** [Presentation and Soundness] The LongMemEval dataset should be tested on other papers' existing baselines (with external systems like RAG). However, no "baseline section" is mentioned in this paper, and I cannot find the baseline mentioned in the "experimental setup" in Section 5.1 as well as in Tables 2 and 3 and their titles (e.g., "K = V is the baseline from xxx paper"). Hence, it is challenging to understand the prior works' baselines based on each individual's assumption. If non-familiar readers try to search for the "baseline" keyword, they will find it only appears once in the second contribution (in Line 95). After jumping to Section 5.3, they cannot find the recent papers conducting LLM with RAG baselines within this section either. If no comparable baselines are suitable for this task, see S2 below for suggestions.

**Weakness 3.** [Presentation and Soundness] **The authors claim this paper presents a *unified* framework, as shown in Figure 5, but without a formal mathematical definition, it is difficult to justify whether this is indeed true.** For example, how does the proposed framework encompass the existing RAG frameworks? Moreover, it is also hard for other future researchers to follow/revise/expand their ongoing work, particularly the four control points (CP) in Figure 5. Take the conducted CoN experiment (Section 5.5) as an example: The connection between the case when CoN is applied and Figure 5 is unclear. This figure is not illuminating, and the imprecise definition will let the readers interpret this paper on their own. Note that each separated component can be visualized and formalized in a better way (see Figure 1 in [0] and [1] for visualization; see also Section 3.1 in [1] for formalization). In addition, this paper only displays general terms without an explicit example to showcase their framework in Figure 5, whereas it is common for various RAG papers (see [2] and [3]).

**Weakness 4.** [Contribution] **As for the dataset comparison, LongMemEval needs justification to significantly distinguish it from other related works, such as task-oriented dialogues (as stated in Line 50) and other QA-based datasets (see C2 below).** Specifically, several question types are highly similar to (that is, can be leveraged by) those existing datasets, especially the single-session scenarios. Nonetheless, they are missing in the references. Hence, further analysis is necessary regarding why not to leverage their datasets.

[0] Tree of Thoughts: Deliberate Problem Solving with Large Language Models, Shunyu Yao, Dian Yu, Jeffrey Zhao, Izhak Shafran, Thomas L. Griffiths, Yuan Cao, Karthik Narasimhan, NIPS 2023

[1] Memory-Based Model Editing at Scale, Eric Mitchell, Charles Lin, Antoine Bosselut, Christopher D. Manning, Chelsea Finn, ICML 2022

[2] Self-RAG: Learning to Retrieve, Generate, and Critique through Self-Reflection, Akari Asai, Zeqiu Wu, Yizhong Wang, Avirup Sil, Hannaneh Hajishirzi, ICLR 2024

[3] REPLUG: Retrieval-Augmented Black-Box Language Models, Weijia Shi, Sewon Min, Michihiro Yasunaga, Minjoon Seo, Richard James, Mike Lewis, Luke Zettlemoyer, Wen-tau Yih, NAACL 2024


# Details

## Presentation (P)

**P1. The contexts need to be reorganized to provide a clear flow.**

* As shown in Figure 2, the dataset construction pipeline is (a) Question Construction $\rightarrow$ (b) Evidence Session Construction $\rightarrow$ (c) History Construction. However, the (a) and (b) steps are merged into a single "paragraph" in Line 241 (within Section 3.2: LongMemEval: Benchmark Curation), while the (c) step has its own single "subsection" (Section 3.3: History Compilation).

* In the main paper, the results and discussion of their proposed framework only have two pages (Sections 5.2 to 5.5), including the two tables and a figure. More analysis should be moved to the main content. For instance, I find Table 7 in Appendix C informative, so perhaps the authors could consider moving this to the main content.

* There is a discrepancy between Section 4.2 and Figure 5. In Section 4.2, the authors use "CP 1: Value" and "CP 2: Key" as the paragraph title. However, it is "CP 1: Key" and "CP 2: Value" in Figure 5.

* What is NL in Figure 7? I think it is natural language (as opposed to JSON format), but "NL" is not a common abbreviation and is not mentioned in the main content either. Moreover, this figure reports the result on LongMemEval_s, yet it is not mentioned in Figure 7 and Section 5.5.

**P2. The figures/tables do not follow the order or ICLR format. Many of them require adjustment in format and location in this paper for better reference.**

* Should follow the ICLR format and place the table title on top (see Tables 1, 2, and 3).
* Tables and Figures are placed on page "i" but first mentioned on page "i+1." For example, Table 1 is first mentioned on page 3 (Line 146) but placed on page 2. For another example, Figure 2 is first mentioned on page 5 (Line 242) but placed on page 4.
* As shown in Figures 2, 3, and 4, it really confuses me to place (a), (b), and (c) at the end. Take Figure 3 as an example:
```
LongMemEval challenges chat assistants through its diverse question types (a), emphases on multi-session reasoning (b),
and diverse evidence locations within sessions (c).
```
It should be:
```
LongMemEval challenges chat assistants through its (a) diverse question types, (b) emphases on multi-session reasoning,
and (c) diverse evidence locations within sessions.
```
* The column separator line is inconsistent (Tables 1 and 3).
* In Table 1, can the authors explain why you do not use "# data" but "# Q"?
* In Line 83, the term session has a different meaning compared to Table 1 (session means the total number of LongMemEval dataset in "with 500 sessions").
* The text in Figure 2 is small. Moreover, the "Answer: 5" is wrong in Question Construction (should be 50).
* In Figure 3-c, the round index 7 ~ 10 is "invisible," compared to the "barely visible" number of sessions = 6 in Figure 3-b. If they do have "height," perhaps insert the numeric values above these bars.
* In Figure 4-b, two "Llama-3.1 8B Instruct" are presented in both w/o CoN and w/ CoN. Which one is 70B?
* Moreover, no explicit evaluation metric (accuracy) is found in Figure 4-b and its caption. While the authors mention "... compared to answering the questions based on only ...," this makes readers need to first observe it is the question answering task, then refer to Section 3.3 to conclude that this table is indeed using accuracy (but not Recall@k and NDCG@k).
* In Figure 5, presenting "key 4," "value chunk 2," etc. without a simple example is hard to grasp the idea. This should be done by showing different color blocks in "legend" (just like Figure 3) and providing a concrete example (possibly re-use the examples in the previous figures). Additionally, Figure 5 does not visualize a "smooth" pipeline; there is a break between the"(1) indexing" process and the "(2) retrieval and (3) reading" process. Specifically, there is no connection between the "key 4 and key 2" and the data storage system.

**P3. Too much emphasis on many terms and two (or more) adjectives to describe certain words.** Specifically, the overuse of "\textit" and compound words (excluding common words in NLP like long-term). While I could understand that the authors want to emphasize the *scenario* they aim to test and *why* not conduct more experiments etc., it further adds the difficulty to grasp the main point in this paper. The texts in italics are already overwhelming even without these. Below are several sentences that may distract the readers:

* In Lines 287-288: This design allows us to simulate *realistic* and *extensible* user-AI chat histories with *minimal conflicts*.
* In Lines 319-320: ... we compare this *online* memory evaluation with *offline reading*, where GPT-4o is prompted to answer with the complete history provided as context. (Note: online is already in italics in Lines 80 and 183)
* In Line 426: Using LongMemEval_m, we compare different value choices in a *budget-aware* manner.
* In Line 430: The only exception is with *multi-session reasoning* questions, where fact decomposition consistently improves performance. (Note: multi-session reasoning is already mentioned and abbreviated as MR before)
* In Lines 476-477: To address this need, we introduce a simple yet effective *time-aware indexing and query expansion* scheme. (Note: "time-aware indexing and query expansion" is already in italics in Line 400; moreover, "time-aware indexing and query expansion" first appears in Line 98, but only time-aware is in italics)
* The authors spend 6 lines mentioning that this paper is inspired by the needle-in-the-haystack test (in Lines 292-297); however, it is already stated in the Introduction Section (in Lines 77-78). In my opinion, the authors should remove this paragraph and elaborate more on either (1) the framework formulation or (2) other experiments that the authors found worth noting in the main content (for instance, Table 7).
* As for the over-use of adjectives/compound words, please refer to Lines 72-79 as an example (others are scattered in the main content, particularly in the Introduction):
```
We introduce LongMemEval, a comprehensive, challenging, and scalable benchmark designed to assess the long-term memory capabilities of chat assistants. LongMemEval consists of 500 human-curated, high-quality questions to test five core memory abilities: (...). Each question requires recalling information hidden within one or more multi-turn task-oriented user-AI dialogues that are LLM-simulated and human-edited. Inspired by the “needle-in-a-haystack” test (Kamradt, 2023), we design an attribute-controlled pipeline to compile a coherent, extensible, and timestamped chat history for each question.
```


## Soundness (S)

**S1. The proposed unified framework is not formalized.** See Weakness 3 above.

**S2. The authors do not compare their proposed framework with other similar baselines such as LLM with RAG or external systems.** While the authors do report 97 data using proprietary LLMs (ChatGPT and Coze) in the pilot study, this subset of data is 5x smaller than the original size of the data (500), not to mention the conversation sessions are extremely short, which is mentioned in Lines 317-318: approximately 10x shorter than LongMemEval_s. What's more, the distribution does not even really match the original LongMemEval_s dataset (see Lines 1030-1034 in Appendix B). Is there at least an implicit baseline comparison between your and other works in this paper in some way (see Weakness 2 above)? Are the works mentioned in Line 393 the baselines?

On the other hand, *if the nature of this dataset is that no prior baselines are suitable for comparison, then the authors should test their framework using more LLMs to benchmark the LongMemEval dataset for future researchers and demonstrate its effectiveness across various LLMs.*
In this scenario, to strengthen the method, it would be necessary to (a) run the experiments multiple times (see [4]) or (b) enlarge the existing LongMemEval dataset. This paper only has 500 data and experiments on two LLMs in its current state: GPT-4o and Llama 3 (70B and 8B). If the budget is an issue, testing other small LLMs (e.g., below 7B) is also a welcome contribution.

**S3. Regarding the soundness in Figure 4-a, the authors only test the closed-source LLMs with a memory system once in a small (97) data.** While the authors already mention (multiple times) that they are interested in testing "*online* interactions with chatbot", this topic further narrows down to a very specific setting. Moreover, as these LLM-based memory systems do not have a snapshot (nor do they release a technical report demonstrating the robustness of their memory systems on various reasoning tasks), including this potentially immature result in the main content could be a problem for future reference (despite the evaluation time mentioned in the footnote), because these systems would be constantly improved over time without any notification, increasing the difficulty in terms of reproducibility.

**S4. The dataset contains only 500, which is rather small. Moreover, many questions focus on *single-turn* sessions (31%; see Figure 3-a).** As these settings are constantly tested in previous long-term/short-term datasets (see Contribution C2 below), the authors should create a dataset with more "multi-session, knowledge update, and temporal reasoning" for current LLMs.

**S5. In Figure 1, the definition of those question types is unclear.** Specifically, in Figure 1, why not treat the "knowledge-update" as a "multi-session" example, and vice versa? Moreover, after adding the time and date in each session, they can also be generalized to the "temporal-reasoning." The authors do not define these types of questions clearly and only use "The other types of questions are multi-session (MR), knowledge-update (KU), and temporal-reasoning (TR)" in Lines 236-237, and yet the distribution of question types is distinctly shown in the pie chart in Figure 3-a.

## Contribution (C)

**C1. Could the authors explain why it is necessary to differentiate "human-AI" and "human-human" conversations?** Am I missing something in this paper? If so, could you kindly explain why you are interested in the user-AI setting and further differentiate this in Table 1? For instance, is there a significant difference between them when training an LLM or in some tasks that prefer user-AI datasets over human-human ones?

**C2. Prior works of task-oriented dialogues (TOD) are missing, such as the MultiWoZ dataset [5].** On the other hand, the AirDialogue dataset [6] has a more narrower scope. **As a result, there may be an issue in Lines 48-50**: *Many datasets focus solely on human-human conversations (Xu et al., 2022a; Maharana et al., 2024; Kim et al., 2024), while others omit task-oriented dialogues, which represent a significant portion of chat assistant usage.* **As for the Related Work, there is another issue in Lines 142-144** (*Despite these advancements, existing QA-based benchmarks overlook several memory capabilities critical to long-term user-assistant interactions: synthesizing information across numerous sessions, recalling assistant-side information, ...*) **because the CoQA dataset [7] encompasses these two issues.** The model needs to refer back to the conversation history. Lastly, **the SituatedQA dataset [8] $-$ related to temporal context and retrieval $-$ is missing in the Related Work.**

[4] Self-consistency improves chain of thought reasoning in language models, Xuezhi Wang, Jason Wei, Dale Schuurmans, Quoc V Le, Ed H. Chi, Sharan Narang, Aakanksha Chowdhery, and Denny Zhou, ICLR 2023

[5] MultiWOZ 2.2 : A Dialogue Dataset with Additional Annotation Corrections and State Tracking Baselines, Xiaoxue Zang, Abhinav Rastogi, Srinivas Sunkara, Raghav Gupta, Jianguo Zhang, Jindong Chen, NLP4ConvAI 2020

[6] AirDialogue: An Environment for Goal-Oriented Dialogue Research, Wei Wei, Quoc Le, Andrew Dai, Jia Li, EMNLP 2018

[7] CoQA: A Conversational Question Answering Challenge, Siva Reddy, Danqi Chen, Christopher D. Manning, TACL 2019

[8] SituatedQA: Incorporating Extra-Linguistic Contexts into QA, Michael J.Q. Zhang, Eunsol Choi, EMNLP 2021

**Questions:**

Please address the concerns/questions in the above presentation (P1-P3), soundness (S1-S5), contribution (C1-C2), and the following other questions.

Other:

1. The Ethics Statement is missing.

* Since the LongMemEval dataset and the pilot study require human annotators, the ethics statement *should* be included, even though they are not as strongly recommended as the Reproducibility Statement. Moreover, given that this paper studies a three-stage framework for improving a chatbot's long-term memory, could this be exploited by malicious users to produce harmful results? I cannot find it in the appendix, however.

2. Do the authors include the Limitations section? If not, then is this work thoroughly studied? What are the future work?

3. What is the temperature used in this paper? What is the version of GPT-3.5 turbo tested in Figure 4-a?

4. In Figure 4-a, do the author have insights regarding why gpt-4o-mini is better than gpt-4o?

5. Do the authors touch upon the NDCG metric in the discussion section?

6. In Lines 419-420, are the timestamps sorted by ascending or descending order? Do the authors evaluate the performance when the timestamps are not sorted in the preliminary experiments?

7. In Table 2, why K = summary and K = V + summary experiments are not included in this paper? Is it because of the budget-aware manner? Moreover, why is the numeric value (0.590) in QA performance using Llama-3 (70B) in bold? It is smaller than the corresponding value when K = V (0.592).

8. The rounding is confusing.

* In the third contribution (Line 99), the authors report improvement is 7% ~ 11%, but it's 6.7% ~ 11.4% in Lines 480-481. Moreover, the result is for GPT-4o LLM, and this improvement does not hold when Llama-3.1 (8B) is used, as stated in Lines 482 (*We also find that the effectiveness of this method depends on using a strong LLM*). Nevertheless, this strong prerequisite is not mentioned in Line 99, which could be a bit misleading regarding the "7% ~ 11%" improvement.

* Can the author provide how to compute the numerical values in Line 467 (*This approach, particularly when using user facts, yielded an average improvement of 4% in retrieval metrics and 5% in final accuracy across all models.*)? Specifically, how to get the result of a 5% improvement in Table 2? Is it $1/3 * [(0.714-0.670)/0.670 + (0.584-0.570)/0.570 + (0.490-0.464)/0.464] = 0.04875$, then 4.9% rounds to 5%? If so, why **only** use Top-5 in GPT-4o and Top-10 in Llama-3.1 (70B and 8B)? Because I can get the 4% result in retrieval metrics if I average all the improvement in "Recall@5, NDCG@5, Recall@10, and NDCG@10": $1/4 * [(0.732-0.706)/0.706 + (0.620-0.617)/0.617 + (0.862-0.783)/0.783 + (0.652-0.638)/0.638] = 0.0411$ (4.1%); however, when I try to average all the top-5 and top-10 improvement in these LLMs (6 entries), it does not match 5%.

* Similar to the previous question, can the authors list the steps when computing the average metric in Lines 480-481? How can we get the 11.4% and 6.7% results from Table 3? Is there any additional rounding method involved after the computation?

9. Are Figure 4-b and Figure 7 related in any sense? If so, please explain the following questions.
* There is a discrepancy between Figure 4-b and Figure 7. Specifically, the oracle of GPT-4o is 0.924 and Llama-3.1 (*70*B, I guess) is 0.848 in Figure 4-b, and they can be found in Figure 7 (JSON + CoN). However, the oracle of Llama-3.1 (8B) is 0.710 in Figure 4-b, which is the natural language with CoN (NL + CoN). Since the JSON input format is not mentioned in Section 3.5, the authors should report when the input format is NL but not JSON.
* In this vein, if I understand correctly, how can we derive the oracle of those LLMs w/o CoN from Figure 7? These values are not identical (I thought it could be done by looking at NL + Direct Prediction).
* Is it a coincidence that Llama-3.1 (8B) has no improvement when CoN is used?

10. Other typos and suggestions:
* To improve the clarity of this paper, including a simple, concrete example in Figure 5 (or mentioned in Sections 4 and 5) would be beneficial. Otherwise, readers must refer to Appendix D (in Figures 11 and 12) to know what keyphrases/user facts/query expansion is exactly.
* The term "round" is not precisely defined in this paper (see Line 377: decomposing sessions into individual rounds). I only find the session is defined in Section 3.1. For instance, can a session be decomposed into 2 (i.e., user and chat assistant) rounds? If more rounds ($>$ 2) are possible, how do you handle this process in this paper?
* cross-session $\rightarrow$ multi-session (for consistency). Note that other words need to be revised for consistency, yet it is too cumbersome to list all of them.
* In Line 65 (verb missing): ... Finally, we the coverage of five core ...
* In Line 138, the em dash is inconsistent.
* In Line 161, RAG is not mentioned in retrieval-augmented generation but is later used in Sections 5.2 and 5.3.
* In Lines 375-376, should use \citep.

**Details Of Ethics Concerns:**

The dataset and pilot study require human annotators but no ethics statement included.

---

> ### Author Response · Authors · 2024-11-24
> **Author Response (Part 1)**
>
> We sincerely appreciate your comprehensive comments and are committed to improving the quality of the paper. Please kindly review the updated draft PDF along with the accompanying responses below.
>
> As your weaknesses section simply summarizes the detailed comments, we directly address the detailed comments point by point.
>
> ---
>
> # **Presentation**
>
> **We have resolved your paper writing comments in the updated PDF (highlighted in orange) and we respond to the other questions and concerns below.**
>
> ## P1 Context reorganization
>
> > Moving more analysis to the main content
>
> Balancing the sections was indeed a hard trade-off. In the submitted version, we decided to emphasize on the benchmark’s quality and its challenging nature. In the revised paper, we have moved Table 7 into the main text such that it can serve as an informative complement to Figure 5.
>
> > What is NL in Figure 7? What is the result reported in Figure 7?
>
> NL in Figure 7 is natural language. We clarify that Figure 7 is measured with “oracle retrieval” with only the evidence sessions. This experimental design isolates the effects of imperfect retrieval. We have revised section 5.5 to further clarify these issues.
>
> ## P2 Figures and tables
> We have improved the content and placement of the figures and tables based on your feedback.
>
> > Why #Q instead of #data in Table 1?
>
> For Table 1, we choose to use #Q because several datasets such as MemoryBank and LoCoMo construct a number of questions from each session. Therefore, it is hard to define a “datapoint”.
>
> > “session” in line 83 has a different meaning
>
> In this paper, “session” always has the same meaning. The “session” in line 83 means that the evidence sessions for each problem in LongMemEval_m are embedded in chat histories that contain 500 sessions. It is a coincidence that LongMemEval also has 500 problems.
>
> > Figure 5 presentation and further examples
>
> The goal of Figure 5 is to illustrate the concept of the memory framework in the most general case. Taking the indexing stage as an example, there are multiple specific designs (facts, entities, time, etc.) for keys and an exponential number of combinations. As a result, providing oversimplified examples would give the readers wrong impressions. In our revision, we have moved the original Table 7 to the main text to further clarify concepts in Figure 5.
>
> ## P3 Emphasis and compound word overuse
>
> We have significantly reduced the use of `\textit` and compound words throughout the paper, especially in the introduction. We hope these modifications improve the paper’s clarity and readability.
>
> ---
>
> # **Soundness**
>
> ## S1 Formulation of the unified framework
>
> We have provided a formulation of the system in section 4.1. Below, we formulate the system with more detailed explanations.
>
> We formulate long-term memory as a massive key-value datastore $D = [(k_1, v_1), (k_2, v_2), ...]$, where the keys $k_i$ can be heterogeneous and the values $v_i$ might repeat. More concretely, a key could be discrete or continuous. In the discrete case, the key could be a sentence, a paragraph, a fact, or an entity, etc. In the continuous case, the key could be e.g., the model’s internal representation under some inputs.
>
> For the three stages, we formulate them with four functions: $\mathcal{I}$, $\mathcal{Q}$, $\mathcal{S}$, $\mathcal{R}$. The offline index function $\mathcal{I}(S, D)$ converts a session $S$ into a series of key-value tuples. Alternatively, the online index function $\mathcal{I}_{online}(S, D)$ updates D with the key-value tuples extracted from S, potentially removing or editing the existing items in D. The query formulation function $\mathcal{Q}(q)$ converts a natural language user query $q$ into a representation $q’$ that function $\mathcal{S}$ could leverage. The salience scoring function $\mathcal{S}(q’, D)$ orders D by their relevance to $q’$. Potential initializations of $\mathcal{S}$ include ranking dense text embedding, graph-based ranking, and so on. Finally, the reading function $\mathcal{R}(M, D’)$ combines the model M with D’, the most relevant part as ranked by the salience scoring function $\mathcal{S}$ and generates a response. The methods outlined in the columns of Table 7 could be seen as different ways to instantiate the four functions $\mathcal{I}$, $\mathcal{Q}$, $\mathcal{S}$, $\mathcal{R}$.

---

> > ### Comment · Reviewer_LhuP · 2024-12-02
> >
> > > For Table 1, we choose to use #Q because several datasets such as MemoryBank and LoCoMo construct a number of questions from each session. Therefore, it is hard to define a “datapoint”.
> >
> > As far as I understand, the common way to report the size of a dataset would be to report the "datapoint" (not solely the # questions), even though there are multiple "conversation sessions" in a really long-term dialogue.
> > If "only" report # Q, it is possible that this column will somehow "inflate" the number of such dataset at first glance.
> > For instance, similar work in [1] reports 3,581 in the released English version. If those 5 "conversation sessions" are treated as one, then it is roughly 700-ish.
> >
> > Moreover, the CoQA and TopiOCQA ([2]) papers both report # passage and # QA pairs in Table 2.
> >
> > > In this paper, “session” always has the same meaning. The “session” in line 83 means that the evidence sessions for each problem in LongMemEval_m are embedded in chat histories that contain 500 sessions. It is a coincidence that LongMemEval also has 500 problems.
> >
> > Thank you for your clarification. I have another question in Table 1: In "# Sess" column, does "50k" in your dataset refer to both LongMemEval S and M?
> >
> > > We have provided a formulation of the system in section 4.1. Below, we formulate the system with more detailed explanations.
> >
> > That sounds legit, though I would prefer to include this in the main content. I am raising this concern because the term "unified" should formalize (from the authors' best of knowledge) every possibility, such as the work in [3] (see Figure 1).
> >
> > Hence, in the initial version of your paper, I would expect something in the three-stage pipeline like $\mathbf{Index}$ x $\mathbf{Retrieve}$ x $\mathbf{Reading}$, and "$\mathbf{Index}$ = \{prior work 1, prior work 2, authors' method 1, authors' method 2, ...\}" (Some of which are from the previous work, while others are newly designed in this paper.)
> >
> > As Table 7 is included in the main content (now Table 2), it is addressed in a different way.
> >
> > [1] Sanghwan Bae, Donghyun Kwak, Soyoung Kang, Min Young Lee, Sungdong Kim, Yuin Jeong, Hyeri Kim, Sang-Woo Lee, Woomyoung Park, Nako Sung, Keep Me Updated! Memory Management in Long-term Conversations, EMNLP Findings 2022
> >
> > [2] Vaibhav Adlakha, Shehzaad Dhuliawala, Kaheer Suleman, Harm de Vries, Siva Reddy, TopiOCQA: Open-domain Conversational Question Answering with Topic Switching, TACL 2022
> >
> > [3] More than Classification: A Unified Framework for Event Temporal Relation Extraction

---

> ### Author Response · Authors · 2024-11-24
> **Author Response (Part 2)**
>
> ## S2.1 Baseline comparison
>
> We first observe that a number of baselines are conceptually captured by the unified framework (Table 7). For example, the standard in-context retrieval augmentation is the case with K = V = session or K = V = round. For another example, the MemoryBank method is the case with K = V = user facts, with natural language format and no CoN for reading.
>
> For this paper, we make the conscious choice to not call these baselines by their names because their out-of-the-box performance is poor. For instance, MemoryBank has nearly zero accuracy on LongMemEval due to its tendency to memorize topics and high-level information instead of user details. In our preliminary experiments, we have to make substantial changes to improve its performance, such as using round as the values, using in-context learning for fact extraction, and reading-stage optimizations. At that point, it would be unfair and confusing to call it by the original name.
>
> Due to these reasons, we opt for independent formulation and implementations to enable controlled comparisons and optimizations of different design factors. We argue that this presentation provides the readers with clearer understanding of the challenges of LongMemEval on different memory components.
>
> ## S2.2 and S4 - Dataset size and question composition
>
> We emphasize that our 500 human-annotated questions are high quality **seed data**. Each question itself could be embedded in an infinite number of chat histories to test the memory ability at arbitrary length. In terms of the size of the expert-curated data, we notice that it is on par with a number of (influential works such as HumanEval ([1], 164 questions), IFEval ([2], 500 prompts), and GPQA ([3], 448 questions). In fact, if one considers the original “needle-in-a-haystack” test, it essentially only contains a few questions, paired with greatly varying context setups. Furthermore, the current dataset size allows us to derive statistically significant results, which is further discussed in the general response.
>
> For the question composition, we argue that comprehensiveness is one of the fundamental goals of LongMemEval. Driven by this goal, we spend a great amount of human effort to propose challenging and diverse questions, as well as balancing different question types. In fact, in the early stage of benchmark construction, we manually removed hundreds of low-quality questions. Each question in LongMemEval could be considered as a full-fledged test that represents a unique scenario with flexibly configurable history.
>
> [1] Evaluating Large Language Models Trained on Code. Chen et al., 2021.
>
> [2] Synthetic Data (Almost) from Scratch: Generalized Instruction Tuning for Language Models. Li et al., 2024.
>
> [3] GPQA: A Graduate-Level Google-Proof Q&A Benchmark. Rein et al., 2023.

---

> > ### Comment · Reviewer_LhuP · 2024-12-02
> >
> > > We emphasize that our 500 human-annotated questions are high quality seed data. Each question itself could be embedded in an infinite number of chat histories to test the memory ability at arbitrary length.
> > In terms of the size of the expert-curated data, we notice that it is on par with a number of (influential works such as HumanEval ([1], 164 questions), IFEval ([2], 500 prompts), and GPQA ([3], 448 questions). In fact, if one considers the original “needle-in-a-haystack” test, it essentially only contains a few questions, paired with greatly varying context setups. Furthermore, the current dataset size allows us to derive statistically significant results, which is further discussed in the general response.
> >
> > > For the question composition, we argue that comprehensiveness is one of the fundamental goals of LongMemEval. Driven by this goal, we spend a great amount of human effort to propose challenging and diverse questions, as well as balancing different question types. In fact, in the early stage of benchmark construction, we manually removed hundreds of low-quality questions. Each question in LongMemEval could be considered as a full-fledged test that represents a unique scenario with flexibly configurable history.
> >
> > It is common that, at least from my prior experience, even if the statistical test is significant, the values of improvement can vary (generally, worse) when enlarging the size of the dataset.
> >
> > This is also related to my other concerns: since other unrelated contexts are inserted in the dataset construction process, why not leverage existing QA datasets in this task to expand the size of LongMemEval? If creating the LongMemEval dataset requires much human effort (and the pilot study), there are many existing information extraction/muti-session/abstention datasets that the authors do not need to construct from scratch.

---

> ### Author Response · Authors · 2024-11-24
> **Author Response (Part 3)**
>
> ## S3 human study
>
> In terms of the analysis, we were indeed limited by the budget. For the setting we presented in Figure 4a, the annotators are required to manually interact with the model round-by-round for multiple sessions. Therefore, analyzing one question for a single model could take more than 10 minutes on average. We further elaborate this challenge in the ethics statement below. In terms of reproducibility, we argue that our result represents best effort under the current LLM landscape with non-transparent API-based (or UI-based) accesses. We also argue that including the results is beneficial because they authentically represent the systems’ performance in a certain time period and sets up an informative reference for future open-source reproductions of the memory systems in the style of ChatGPT and Coze. Our comparison with offline reading also highlights the degree of challenges of building robust online memory systems, even for commercial products.
>
> Based on your suggestion, we also extended the analysis to five smaller LLMs and present the results in the table below. We mainly evaluate the either long-context (LC) generation or generation with top-10 memory retrieval.
>
> Overall, we observe that the performance is consistent with the results reported in the paper. For instance, long-context performance on LongMemEval_S is significantly lower than the oracle retrieval performance. In addition, the recommended fact-based key expansion and CoN consistently improves the performance.
>
> | Model                                | Evidence Session Only | Evidence Session Only | LongMemEval_S | LongMemEval_S | LongMemEval_S | LongMemEval_S | LongMemEval_M | LongMemEval_M |
> |--------------------------------------|-----------------------|---------------------|---------------------|---------------------|----------------------------------------|----------------------------------------|---------------------|---------------------|
> |                                      | LC                    | LC CoN             | LC Direct           | LC CoN             | K = V = round, CoN            | K = V+fact, V = round, CoN     | K = V = round, CoN            | K = V+fact, V = round, CoN     |
> | mistralai/Mistral-Nemo-Instruct-2407 | 0.688                 | 0.686              | 0.162               | 0.286              | 0.640                                  | 0.666                                 | 0.554                                 | 0.598                                 |
> | Qwen/Qwen2.5-7B                      | 0.282                 | 0.504              | 0.128               | 0.144              | 0.452                                 | 0.462                                 | 0.390                                  | 0.424                                 |
> | microsoft/Phi-3.5-mini-instruct      | 0.488                 | 0.490               | 0.312               | 0.352              | 0.550                                  | 0.570                                  | 0.462                                 | 0.468                                 |
> | Llama-3.2-3B-Instruct                     | 0.522                 | 0.636              | 0.008               | 0.010               | 0.466                                 | 0.508                                 | 0.466                                 | 0.470                                  |
> | Llama-3.2-1B-Instruct                     | 0.386                 | 0.398              | 0.010                | 0.016              | 0.312                                 | 0.336                                 | 0.286                                 | 0.312                                 |
>
> ## S5 Question type definition
>
> As defined in section 3.1, each instance is a 4-tuple (S, q, t_q, a). As such, we differentiate the task types not only based on S, but also based on their (q, a) distribution. Our final ontology ensures that all the instances of the same type have the same (S, q, t_q, a) distribution and format.
>
> Specifically, for “knowledge-update”, the sessions either explicitly define the user’s old life state, or explicitly verbalize the new life state, or both. On the other hand, the “multi-session” questions are mainly aggregation or comparison questions that require the model to collect supporting facts at multiple places.
>
> For “temporal-reasoning” questions, the (q, a) pairs are constructed such that the answers strongly depend on the timestamps of both t_q and the timestamps of the evidence sessions. For all the tasks that are not “temporal-reasoning”, we ensure that the answer is time-independent, i.e., the question could always be reliably answered with any set of arbitrary timestamps.
>
> In our final revision, we will add a table to clarify how the task types are defined. This will also be clarified more when the readers are able to see the full released dataset.

---

> > ### Comment · Reviewer_LhuP · 2024-12-02
> >
> > > As defined in section 3.1, each instance is a 4-tuple (S, q, t_q, a). As such, we differentiate the task types not only based on S, but also based on their (q, a) distribution. Our final ontology ensures that all the instances of the same type have the same (S, q, t_q, a) distribution and format.
> >
> > > Specifically, for “knowledge-update”, the sessions either explicitly define the user’s old life state, or explicitly verbalize the new life state, or both. On the other hand, the “multi-session” questions are mainly aggregation or comparison questions that require the model to collect supporting facts at multiple places.
> >
> > > For “temporal-reasoning” questions, the (q, a) pairs are constructed such that the answers strongly depend on the timestamps of both t_q and the timestamps of the evidence sessions. For all the tasks that are not “temporal-reasoning”, we ensure that the answer is time-independent, i.e., the question could always be reliably answered with any set of arbitrary timestamps.
> >
> > > In our final revision, we will add a table to clarify how the task types are defined. This will also be clarified more when the readers are able to see the full released dataset.
> >
> > This still does not address my concern, for common knowledge update, it would be related to the real-world fact like a change in the president (though it not my main point here).
> >
> > I would like to emphasize that in your examples of Figure 1, the user first stated that he had three bikes, then states that he got another bike in the next session. This "knowledge-update" can be treated as "multi-session" (aggregating supporting facts at multiple places).
> > In your explanation, it would be something like
> > * 1st session: ``user state = three bikes``
> > * 2nd session: ``user state = four bikes``  (= 1st knowledge update)
> >
> > Similarly, in "multi-turn session," the user mentioned she got four instrument in four sessions, then why not regard this data instance as "knowledge-update"? For instance:
> > * 1st session: ``user state = 1 instrument``
> > * 2nd session: ``user state = 2 instruments`` (= 1st knowledge update)
> > * 3rd session: ``user state = 3 instruments`` (= 2nd knowledge update)
> > * 4st session: ``user state = 4 instruments`` (= 3rd knowledge update)
> >
> > There isn't really a difference in your definition.

---

> ### Author Response · Authors · 2024-11-24
> **Author Response (Part 4)**
>
> ---
>
> # **Contribution**
>
> ## C1 Human-AI vs. Human-Human
>
> Thank you for raising this question. We argue that there is a fundamental difference between building systems for dialogue modeling (human-human chat) and chat assistants (human-AI chat).  Specifically, two stylistic aspects are different: topic distribution and information density.
>
> **Topic Distribution.** For human-human chat, each side of the conversation has a personality. The conversation is driven by social interaction, information exchange, or relationship building. On the other hand, user-assistant chats are more likely to be information-seeking. In terms of the style, the information in human-human chats is also likely to be presented in a more colloquial style. Therefore, the two types of conversation represent significantly different distributions, which pose challenges even for language modeling.
>
> **Information Density.** More importantly, human-AI chats challenge long-term memory with a lower density of personal information. For various chat assistant tasks, the user may provide long-context inputs, and the assistant can provide long-formed responses. In such scenarios, it is nontrivial for the memory system to decide what to memorize and how to organize the memory structure for more effective retrieval. This problem is more salient for agentic tasks, where the assistant is often provided with large observation contexts.
>
>
> ## C2 Prior work on task-oriented dialogues
>
> Thank you for the suggestions. **We have revised the introduction and the related work to clarify this and cite previous works on task-oriented dialogues you have mentioned.**
>
> We would like to note that the major prior works are the papers on long-term conversational memory, and our previous draft assumed it in a number of locations. Datasets such as MultiWoZ, AirDialogue, CoQA, and SituatedQA do not evaluate chat assistants on the long-term memorization abilities with long-term dependencies between the sessions, which is the core of LongMemEval.
>
> Specifically, SituatedQA evaluates question answering within a given temporal or situational context. Its challenges come from pinpointing the correct version of public knowledge, instead of organizing the user information during user-AI interaction. For CoQA, it evaluates dialogue understanding across multiple turns in the same session, and its history length is also very short by the current standard. Therefore, we argue that these datasets are less relevant than the related works discussed in the paper such as LoCoMo and PerLTQA.
>
> In comparison, LongMemEval designs a unique method to embed personal details in extensible task-oriented dialogue histories without causing conflicts. We encourage the readers to envision LongMemEval as the counterpart to the “needle-in-a-haystack” test for long-term memory abilities, which is unique among the current literature.

---

> > ### Comment · Reviewer_LhuP · 2024-12-02
> >
> > > Thank you for raising this question. We argue that there is a fundamental difference between building systems for dialogue modeling (human-human chat) and chat assistants (human-AI chat). Specifically, two stylistic aspects are different: topic distribution and information density.
> >
> > > Topic Distribution. For human-human chat, each side of the conversation has a personality. The conversation is driven by social interaction, information exchange, or relationship building. On the other hand, user-assistant chats are more likely to be information-seeking. In terms of the style, the information in human-human chats is also likely to be presented in a more colloquial style. Therefore, the two types of conversation represent significantly different distributions, which pose challenges even for language modeling.
> >
> > > Information Density. More importantly, human-AI chats challenge long-term memory with a lower density of personal information. For various chat assistant tasks, the user may provide long-context inputs, and the assistant can provide long-formed responses. In such scenarios, it is nontrivial for the memory system to decide what to memorize and how to organize the memory structure for more effective retrieval. This problem is more salient for agentic tasks, where the assistant is often provided with large observation contexts.
> >
> > This is reasonable, but I am more of asking whether there are any existing works that further differentiate this from the authors' knowledge (such as in training chatbot, etc.).

---

> ### Author Response · Authors · 2024-11-24
> **Author Response (Part 5)**
>
> # **Other Questions**
>
> > Q1 Ethics statement
>
> We have added an ethics statement section at the end of the paper. In the ethics statement, we discuss the human annotation standards we uphold and potential societal effects of the LongMemEval.
>
> > Q2 Limitations
>
> Beyond the robustness issues mentioned in the newly added ethics statement, we also recognize limitations in both evaluation and modeling. On the evaluation side, further work could benefit from (1) introducing more complex multi-hop temporal reasoning questions, such as querying the contents in some session using another session and the relative time as the reference; (2) consider long-term evolution of the user’s relationships between other users; and (3) directly evaluate user satisfaction in the personalized chat. These scenarios pose a harder challenge than LongMemEval and would require significant additional efforts to accomplish. On the modeling side, future work would benefit from a fully online memory system that automatically resolves the knowledge conflicts during interaction. For these systems that read and write for every session, efficiency is also a significant concern, as the system would benefit from automatically deciding when and what to read/write.
>
> We have incorporated this discussion into the updated draft (Appendix Section F).
>
> > Q3 Temperature and model version
>
> We set the temperature to 0 for all the QA experiments. The GPT-3.5 version is gpt-3.5-turbo-0125 for Figure 4a.
>
> > Q4 Figure 4a
>
> Yes, we have done a number of manual analyses after the human study. We find that the system tends to aggressively rewrite and remove memory items when GPT-4o is used as the model. After several sessions, a number of important details are likely to be wiped out. By contrast, we found GPT-4o-mini rewrites and merges fewer memory items, which causes more detail to be retained. We also found that for the human study we conducted, the performance bottleneck is the memory write instead of reading. In other words, as long as the model is able to retain the evidence statement in the memory, it is likely to answer the question correctly.
>
> > Q5 NDCG
>
> Overall, we have found a high correlation between NDCG@k and Recall@k, with a few disagreements when the two metrics are used to evaluate query expansion methods in Table 3. This is mainly because the introduced time-based query manipulates the order using the timestamp metadata. In our implementation, the values with timestamps outside of the specified range will be moved to the last position, causing a low NDCG score. In this case, recall is the more informative metric. We will add a more detailed discussion in the final revision.

---

> ### Author Response · Authors · 2024-11-24
> **Author Response (Part 6)**
>
> > Q6 Timestamp sorting
>
> We sort the sessions by the ascending order of their time (i.e., older sessions come first). We have done preliminary analyses on Llama-3.1 8B Instruct and found that sorting by the time order outperforms using random timestamp order or retrieval order. We thus stick with the practice for the entire paper as this method also presents the contexts in a logical order. This design is crucial when round is used as the granularity as it ensures that items from the same round are placed together.
>
> > Q7 Table 2 K = summary setting
>
> No, we did not include K = summary under Value = Round in Table 2 because the user message in each round is succinct enough such that further summarizing is not required. We have fixed the wrong boldface in the table.
>
> > Q8 Confusing rounding
>
> Thank you for identifying the ambiguous rounding issues.
>
> 11.4% is obtained from [(0.451-0.421)/0.421 + (0.495-0.499)/0.499 + (0.526-0.489)/0.489 + (0.722-0.550)/0.550] / 4. This is the gain in recall for value = round across different key settings and top-k settings. We were using Google Sheets for the average and it might produce some rounding discrepancies. We have modified it to 11.3% in the latest version.
>
> Similarly, 6.7% is obtained from [(0.654-0.639)/0.639 + (0.707-0.651)/0.651 + (0.722-0.684)/0.684 + (0.797/0.721)/0.721] / 4. We have modified it to 6.8% in the latest version.
>
> For the 4% improvement in retrieval metrics, we apologize that we used a wrong row to calculate the averaged result. The actual number should be [(0.644-0.582)/0.582 +(0.784-0.692)/0.692 + (0.732-0.706)/0.706 + (0.862-0.783)/0.783 / 4 = 9.4%
>
> For the 5% improvement in QA accuracy, the more precise result should be [(0.657-0.615)/0.615 + (0.720-0.670)/0.670 + (0.638-0.6)/0.6 + (0.682-0.624)/0.624 + (0.566-0.518)/0.518 + (0.572-0.534)/0.534 + (0.714-0.67)/0.67 + (0.7-0.676)/0.676 + (0.588-0.592)/0.592 + (0.584-0.57)/0.57 + (0.53-0.524)/0.524 + (0.49-0.464)/0.464] / 12 = 5.4%.
>
> We have revised the paper PDF accordingly.
>
> > Q9 Figure 4b and Figure 7
>
> The results in Figure 4b and Figure 7 are not directly comparable. Specifically, Figure 4b uses LongMemEval_S while Figure 7 uses oracle retrieval (only the evidence sessions; explained in the previous replies and clarified in the revised paper).
>
> For Llama-3.1 8B, we observe that CoN is in fact helpful when the memory items are provided in JSON. When the memory items are in natural language, CoN does not improve its performance. We hypothesize that the 8B model has limited ability and confuses the history with the current chat context when NL is used, thus failing to extract the notes effectively. This further suggests the importance of using JSON format to present the memory items.
>
> > Q10 Other typos
>
> Thank you for these editorial suggestions. We have reflected them in the revised PDF. For the requested example, we will instead provide a discussion and combine Table 7 with Figure 5 in the final version (page 7).

---

> > ### Comment · Reviewer_LhuP · 2024-12-02
> >
> > > No, we did not include K = summary under Value = Round in Table 2 because the user message in each round is succinct enough such that further summarizing is not required. We have fixed the wrong boldface in the table.
> >
> > 0.590 is still boldface in Table 3?
> >
> > Lastly, while most of the presentation issues have been resolved, why are some papers cited twice in References?
> > * "REPLUG: retrieval-augmented black-box language models" on page 14
> > * "Augmenting language models with long-term memory" on page 15

---

> ### Author Response · Authors · 2024-11-28
> **General Response to Weaknesses**
>
> Dear Reviewer,
>
> Thank you again for your efforts in reviewing this paper. We would also like to respond to the general weaknesses you raised to ensure that we are aligned with the big picture, although we have responded to the detailed points in the previous response.
>
> > ### Paper Presentation and Proofreading
>
> We have thoroughly revised the paper and fixed the presentation issues such as table/figure arrangement, emphasis or compound word overuse, and rounding issues. We have also improved the content arrangement to enhance clarity, such as arranging the section contents in 3 and bringing in the original Table 7 (now Table 2) to the main text. Finally, we have added an ethics statement (after conclusion), a limitations section (section F), and statistical testing results (in the general response).
>
> Balancing the sections was indeed a hard trade-off. We argue that as LongMemEval is a new challenge, it is important to emphasize the logic behind the proposed testing and justify its difficulty as well as quality. We therefore allocate five pages for this part. Meanwhile, instead of two pages, our empirical study actually spans page 6-10 (starting from section 3.4). We have presented comprehensive results in each subsection, along with highlighted discussions of the most important takeaways. While the paper’s main text is self-contained, we also present further results and analyses in Appendix B, D, and E for interested readers.
>
> > ### Baseline Coverage
>
> As this paper is a resource and evaluations paper, there is no explicit “baseline” to compare with. As explained in the response to S2.1, the memory system studied in this paper subsumes previous methods such as RAG, CoN, and MemoryBank (further explained in Table 2 in the revised PDF). We intentionally avoided explicitly naming these methods because directly applying them results in nonsensical performance. They have to be modified substantially to achieve non-trivial performance on LongMemEval.
>
> > ### Framework Uniformity
>
> We have provided a detailed formulation in our responses to S1, and we have added Table 2 to clarify the discussions in Figure 5. We note that uniformity means that various memory systems could be viewed as specific instantiations of the framework, as shown in the Table 2. We have also added an example in the general response to further illustrate our framework.
>
> > ### Contribution
>
> We would like to highlight that LongMemEval is a novel and challenging benchmark for *long-term memory* of chat assistants. LongMemEval’s novelty lies in a combination of (1) diverse questions that comprehensively cover chat assistant memory abilities such as knowledge update, abstention, and temporal reasoning and (2) long and freely extensible chat history without invalidating the questions. These two features distinguish LongMemEval from other benchmarks such as MultiWOZ, AirDialogue, CoQA, and SituatedQA.
>
> We provided a more detailed comparison in our previous response and further highlighted the comparison with the paper you suggested in our revised PDF to ensure clarity. We believe LongMemEval is distinct from those works and has a unique contribution to the field.
>
> ---
>
> If you have additional questions that we can further clarify, please kindly let us know. Thank you!

---

> ### Author Response · Authors · 2024-12-03
> **Further Clarifications**
>
> Dear Reviewer,
>
> We are glad that our previous response seemed to have resolved most of your concerns. We address your follow-up questions below.
>
> > Number of questions vs. data points
>
> In LongMemEval, #Q is smaller than #data points (contrary to your intuition), because each question has its own unique set of evidence sessions and LongMemEval supports constructing a large number of test instances (i.e., “data points”) by combining the evidence sessions with other history sessions. In our evaluation, LongMemEval_S and LongMemEval_M each contain 500 data points. Reporting the number of data points could actually risk inflating the dataset size, and #Q is a more modest way to present the dataset sizes for LongMemEval.
>
> > “50k” sessions in Table 1
>
> In Table 1, “50k sessions” refers to the pool of unique evidence sessions and other generated history sessions. The history sessions used in LongMemEval S and M are sampled from this pool. Since we release the entire LongMemEval pipeline as a resource, we use 50k as the total number of sessions. In practice, LongMemEval_M samples roughly 500*500 = 250k times from the pool and covers roughly 95% of these sessions.
>
> > Unified memory framework formulation
>
> Thank you for recognizing the validity of the definition and that moving Table 7 to the main text clarifies the uniformity concerns. We will combine the definition with Table 2 in the main text in our final revision.
>
> > Dataset size and composition
>
> LongMemEval prioritizes the quality of the questions instead of the size. Compared to previous work, LongMemEval has a uniquely comprehensive memory ability coverage, and extensive efforts are spent to ensure (1) information-dense user messages and (2) the integrity of the question even with many history sessions. We handle these challenges through the novel attribute-controlled data creation pipeline that generates attribute-specific user backgrounds. Extra human efforts are also required to ensure the integrity of time mentioned in all evidence sessions. These features are unparalleled by existing QA/IE literature. Partially reusing previous datasets is also prone to question inconsistency, potential test data leakage, inheriting the bias from previous works as well.
>
> As LongMemEval is already comprehensive, difficult, and enabling statistically significant comparisons, we argue that further including questions of potentially lower quality harms the *democratization* of evaluation. In fact, obtaining a single test result on LongMemEval_M already requires 1.5M * 500 = 750M input tokens, which costs 1250 USD for a memory system purely based on GPT-4o. The cost is ~150 USD for LongMemEval_S. We argue that the current size strikes a good balance between evaluation accuracy and the viability for everyone in the community to use.
>
> > Knowledge update vs. multi-session questions
>
> The major reason for keeping the two question types distinct is that they are created from distinct metadata schemas, which lead to distinct question distributions. Specifically, the metadata for knowledge update is a pair of facts (S, R, O) -> (S, R’, O’) where R’ or O’ or both is modified to invalidate the original R or O. Here (S, R, O) represents (subject, object, relation). In contrast, the metadata for multi-session questions is a list of facts (S, R, O_1), (S, R, O_2), …, (S, R, O_n).
>
> We admit that the two examples presented in Figure 1 accidentally represent the overlap between the two question type distributions, which is infrequent in the full LongMemEval dataset. We will replace the example in Figure 1 to avoid confusions. For your reference, we provide two knowledge update questions that do not fall into this overlap:
>
> Example 1 (R changes while O remains the same)
>
> * Previous fact: my mom finds my grocery list method too difficult to use.
> * New fact: my mom is using the same grocery list method as me.
>
> Example 2 (both R and O change)
>
> * Previous fact: I see my therapist Dr. Smith every two weeks.
> * New fact: I see my new therapist Dr. Brown every week.
>
> > User-AI chat literature
>
> We are aware of several works on using user-assistant chat data for training better chat assistants [1-3]. These works are marginally relevant to this paper’s theme of long-term memory, but illustrate the wide applicability of the user-AI chat format.
>
> [1] OpenAssistant Conversations - Democratizing Large Language Model Alignment. Köpf et al., 2023.
>
> [2] Enhancing Chat Language Models by Scaling High-quality Instructional Conversations. Ding et al., 2023.
>
> [3] LMSYS-Chat-1M: A Large-Scale Real-World LLM Conversation Dataset. Zheng et al., 2023.
>
> > Presentation issues
>
> Thank you for recognizing that most of the presentation issues have been solved. We apologize for the typo in Table 3 and the duplicated bibtex entries. We will fix these minor issues in the final revision.
>
> We hope our response further clarifies your questions, and please let us know if you have any other concerns.
>
> Thank you,
>
> All Authors

---

### Official Review · Reviewer_GckN · 2024-10-26

**Soundness:** 3
**Presentation:** 3
**Contribution:** 3
**Rating:** 6
**Confidence:** 4

**Summary:**

This paper focuses on the problem of long-term dialogues. To study this problem, this work constructs LongMemEval, a new benchmark for evaluating five long-term memory capabilities of chat assistants, including information extraction, multi-session reasoning, temporal reasoning, knowledge updates, and abstention.  Experimental results on this new benchmark indicate significant challenges in existing LLMs on long-term dialogues. Furthermore, the authors propose a new framework, consisting of several designs, such as session decomposition, fact-augmented key expansion, and time-aware query expansion. Further results show that these designs can improve both memory recall and downstream question answering.

**Strengths:**

1. Long-term dialogue is an important problem in the era of LLMs, which also lacks of comprehensive problem formulations. This work formulates several core memory capabilities in this problem.
2. Construct a new challenging benchmark for studying this problem.
3. Propose a unified framework for improving the long-term dialogue capabilities in LLM-based conversational systems.

**Weaknesses:**

1. The construction of history sessions is not well motivated. The constructed history could be very noisy and the temporal order is just random. Not sure what is the motivation behind this messy historical session construction.
2. The proposed framework is actually a combination of several existing design.
3. There are too many terms to remember in this work, which hampers the readability of this paper.
4. It would be better to include some long-term memory models to make a comparison for better presenting the challenge of the proposed benchmark and the effectiveness of the proposed framework.

**Questions:**

1. Why is the proposed framework called a "unified" framework? It is very pipeline, not unified.
2. Since the timestamp is randomly assigned, how to guarantee the evaluation of temporal reasoning?

---

> ### Author Response · Authors · 2024-11-24
> **Author Response**
>
> Thank you for acknowledging our effort to formulate and evaluate the long-term memory challenge. We address your insightful questions below.
>
> > Chat history quality
>
> The goal of our history construction process is to simulate diverse yet consistent chats between a specific user and a chat assistant. To this end, we decompose a user’s persona into a number of attributes, and simulate task-oriented sessions along each attribute. During the history construction, beyond the evidence sessions, we draw the rest of sessions from (1) sessions with unique attributes and (2) public task-oriented chats that generally do not reflect user persona. We conducted a preliminary manual analysis of 50 generated histories and found that the pipeline can consistently produce a coherent history. More importantly, this pipeline minimizes the chance of any history session conflicting the information in the evidence sessions, which suffices for the integrity of the problem.
>
> > Timestamp quality
>
> We uphold a high standard for timestamp consistency. Our annotation has a specific timestamp annotation step where we manually inspect all the questions. For all the questions that involve temporal reasoning or temporal mentions, we assign a fixed timestamp to each of the evidence sessions and the question. During history construction, the other sessions are randomly assigned with timestamps such that they interleave with the evidence sessions. This ensures the questions are always correctly answerable.
>
> On the other hand, we ensure that the answers to other questions are timestamp-independent. Logically, they are answerable with arbitrary timestamp assignments. In practice, these questions in LongMemEval_S and LongMemEval_M have their chat history spanning within only three months to minimize potential confusions.
>
> We will highlight these efforts in the final revision of the paper.
>
> > Uniformity of the proposed memory framework
>
> Our framework is indeed not a monolithic architecture, but its uniformity stems from the ability to represent different types of memory-augmented chat systems. We outline the representation in Appendix Table 7. We have also included a formal definition in the response to reviewer LhuP to clarify the framework.
>
> We argue that this framework allows us to have a clear view of the important design choices made by current literature and enables us to design controlled experiments on their effectiveness.
>
> > Long-term memory models
>
> Thank you for this suggestion. As the major contributions of the paper are benchmarking and basic evaluations, we leave the investigation of advanced memory architectures to the future work. However, our memory evaluations are still general enough that informs a number of RAG-based memory models such as MemoryBank [1]. During the rebuttal period, we have also extended our evaluation to more LLMs. Please find the results in the general response.
>
> [1] MemoryBank: Enhancing Large Language Models with Long-Term Memory. Zhong et al., 2023.

---

> ### Comment · Reviewer_GckN · 2024-11-25
>
> Thanks for the reply. I am still leaning positive towards this work. Good luck.

---

### Official Review · Reviewer_eitt · 2024-10-29

**Soundness:** 2
**Presentation:** 2
**Contribution:** 3
**Rating:** 8
**Confidence:** 4

**Summary:**

This paper evaluates long-term memory abilities in dialogue tasks from extraction, multi-session reasoning, temporal ability, knowledge management, and abstention. They found that it's a challenge for existing LLMs to perform sustained interaction with those chat histories. To tackle this issue, this paper presents a unified framework to redefine the process when treating long-term memory. Beyond that, extensive experiments show optimizing in the new proposed framework can improve memory recall and QA ability in their benchmark. The key point of this work is to divide the pipeline workflow into single parts when facing long-term memory dialogue situations. They create larger data to make the long-term memory better represented, then make the model perform better.

**Strengths:**

1. It divides the memory workflow into three phases: indexing, retrieval, and reading. This segmentation not only helps to conceptualize the entire process but also lays the groundwork for systematically improving each stage. Then this paper further enhances this workflow. These modifications represent a thoughtful approach to refining how information is indexed, retrieved, and read from long-term memory. The process is sound and convincing.

2. This paper did a comprehensive evaluation across different existing models and provided detailed insights into how various models perform differently depending on the task—some models excel at reading (i.e., interpreting memory), while others are better at understanding or processing the retrieved information.

**Weaknesses:**

1. This work was generated by human-AI talk. However they did not mention how they control the hallucination and other issues when talking to AI chatbots (i.e., repeating problems, proportion of effective dialogue turns, etc.)

2. The proposed five aspects to evaluate long-term memory should be discussed more (Which aspect contributes more? Do different aspects have overlaps? What if one aspect performs better but another one fails?).

**Questions:**

Please give feedback to Weaknesses -> No.1 and No.2.

---

> ### Author Response · Authors · 2024-11-24
> **Author Response**
>
> Thank you for acknowledging our work’s evaluation effort and raising insightful questions.
>
> > Quality control for human-AI talk
>
> We argue that for LongMemEval to be effective, the core qualities to ensure are that (1) the questions, answers, and timestamps are logically correct and (2) the evidence statements are correctly reflected in the evidence sessions. We spend significant human efforts on each and every question to ensure these two properties. We have implemented a check to detect repetitions and stop the session simulation accordingly. However, we do not have other sophisticated quality controls, as it is beneficial to evaluate the chat assistants in real-world scenarios where these natural artifacts could occur.
>
> > Relationships between the considered aspects of memory ability
>
> Thank you for raising this great question. Among the five abilities we evaluate, we find that information extraction (IE) is the fundamental ability. In fact, it covers all the questions in LongMemEval. Beyond IE, the other abilities have disjoint questions to represent them. We believe that these four abilities represent independent abilities that are all crucial to successful chat systems. In the table below, we analyze the performance by skills of GPT-4o and Llama-3.1 8B Instruct. LC = long-context. For the optimizations, we use top-10 round-level retrieval with fact expansion for key indexing, time-aware query expansion, as well as chain-of-note for memory reading.
>
> | Model                     | Evaluation                          | IE (single session) | MR   | KU   | TR   | ABS   |
> |---------------------------|--------------------------------------|---------------------|-------|-------|-------|-------|
> | **GPT-4o**               | LongMemEval_S LC, CoN               | 0.873               | 0.451 | 0.769 | 0.624 | 0.700 |
> |                           | LongMemEval_S LC, all optimizations | 0.968               | 0.752 | 0.859 | 0.797 | 0.633 |
> |                           | LongMemEval_M, all optimizations    | 0.857               | 0.609 | 0.808 | 0.662 | 0.700 |
> | **Llama-3.1 8B Instruct** | LongMemEval_S LC, CoN               | 0.667               | 0.301 | 0.615 | 0.271 | 0.600 |
> |                           | LongMemEval_S LC, all optimizations | 0.818               | 0.444 | 0.615 | 0.414 | 0.433 |
> |                           | LongMemEval_M, all optimizations    | 0.770               | 0.451 | 0.590 | 0.301 | 0.700 |
>
>
> As shown in the table, we find that while our optimizations significantly improve the performance, the abilities other than single-session IE have large rooms for improvement. For GPT-4o, knowledge update and abstention are the hardest abilities to improve. One potential reason is that it is tempting to generate a plausible answer for these questions based on partially retrieved knowledge. For the Llama-3.1 8B Instruction model, its multi-session reasoning ability limits the performance even with high quality memory retrieval. Finally, we also remark that under the LLM may be able to correctly abstain under poor memory retrieval, which complicates the evaluation and motivates us to propose evaluating the retrieval stage separately from generation.

---

### Official Review · Reviewer_NGvs · 2024-11-04

**Soundness:** 3
**Presentation:** 3
**Contribution:** 3
**Rating:** 8
**Confidence:** 4

**Summary:**

the paper proposes LONGMEMEVAL, a comprehensive benchmark designed to evaluate five core long-term memory abilities of chat assistants: information extraction, multi-session reasoning, temporal reasoning, knowledge updates, and abstention. Based on that, the present a unified framework that breaks down the long-term memory design into four design choices across the indexing, retrieval, and reading stages. Built upon key experimental insights, they propose several memory designs including session decomposition for optimizing value granularity, fact-augmented key expansion for enhancing the index structure, and time-aware query expansion for refining the search scope. Generally, i think this is a good paper and demonstrates some insightful observations via comprehensive experiments.

**Strengths:**

1. the proposed benchmark is valuable, being diverse, scalable and practical.
2. the experimental results are solid and comprehensive and the analysis is insightful such as the choices of key, value and query expansion.
3. the presentation and organization is relatively clear and easy-to-understand.

**Weaknesses:**

1. lots of human intervention is required to ensure the quality of dataset, such as line 248
2. most of experiments are conducted using gpt4o and llama3.1 8b instruct models. there is no comprehensive analysis about exsiting models.

**Questions:**

1. according to (c) in figure 3, it seems there are no answer located in 7 8 9 10 round.
2. in figure6, what motivate you to list multi-session subset independently? and why not analyze this in later experiemnts?
3. why some experiments use both llama 3.1 70b and 8b instruction model, and some only use 8b model?

other typos

1. the answer in (a) of figure 2 is wrong?
2. best performance of llama 3.1 70b when value = session on table 2 is wrong?
3. typo in line 468

---

> ### Author Response · Authors · 2024-11-24
> **Author Response**
>
> Thank you for your great attention to the paper and insightful review.
>
> > Human interventions
>
> We recognize that benchmark involves a lot of human intervention. However, we encourage the readers to view this as an advantage instead of a limitation, as this is the standard approach taken by numerous influential works (such as [1]). In constructing LongMemEval, we ensure that (1) all the questions are correctly answerable; (2) the evidence statements are clear and correct; and (3) the timestamps are logically correct for the related question. These crucial qualities are extremely difficult to guarantee without human interventions.
>
>
> [1] Evaluating Large Language Models Trained on Code. Chen et al., 2021.
>
> > Multi-session specially considered in Figure 6
>
> In Figure 6, we highlighted multi-session because the preference on the granularity is inconsistent with all the other tasks and the averaged performance. However, its preferences for the other design choices align well with most of the other tasks as well as the performance average. Therefore, we treat value granularity as an intriguing case and take special care to mention the results of multi-session.
>
> > Benchmarked models
>
> We note that for all the QA experiments we always use three models as the reader model: GPT-4o, Llama-3.1 8B Instruct, and Llama-3.1 70B Instruct. These models represent the strongest models on the market of their sizes. For the retrieval-related experiments, we mainly used Llama-3.1 8B Instruct as we did not find significant differences between the 8B and 70B version and 8B is better efficiency-wise. We found GPT-4o performs much better than Llama-3.1 8B Instruct only for query expansion and thus discussed its results.
>
> During the rebuttal period, we have also extended our evaluation to more LLMs as the reader models. Please find the results in the general response.
>
> > Writing issues
>
> Thank you for identifying the issues and we have fixed them in the revised PDF, including Figure 2a, Figure 3, Table 2, and line 468.

---

### Author Response · Authors · 2024-11-24
**General Response to Reviewers**

Dear reviewers,

We sincerely thank you for paying great attention to our paper and raising valuable comments. We are glad that all the reviewers recognized the paper’s contribution as “good” with a rating of 3, finding that the paper

* presents a timely and challenging benchmark on the long-term memory issue (NGvs, GckN, LhuP)
* introduces a useful memory workflow framework for a comprehensive evaluation (NGvs, eitt, GckN).
* backs up the argument with solid empirical results (NGvs, eitt)

During the rebuttal, we have added several supporting results.

---

### (1) Results on more reader LLMs

We extend the analysis to five smaller LLMs and present the results in the table below. We mainly evaluate the either long-context (LC) generation or generation with top-10 memory retrieval.

Overall, we observe that the performance is consistent with the results reported in the paper. For instance, long-context performance on LongMemEval_S is significantly lower than the oracle retrieval performance. In addition, the recommended fact-based key expansion and CoN consistently improves the performance.

| Model                                | Evidence Session Only | Evidence Session Only | LongMemEval_S | LongMemEval_S | LongMemEval_S | LongMemEval_S | LongMemEval_M | LongMemEval_M |
|--------------------------------------|-----------------------|---------------------|---------------------|---------------------|----------------------------------------|----------------------------------------|---------------------|---------------------|
|                                      | LC                    | LC CoN             | LC Direct           | LC CoN             | K = V = round, CoN            | K = V+fact, V = round, CoN     | K = V = round, CoN            | K = V+fact, V = round, CoN     |
| mistralai/Mistral-Nemo-Instruct-2407 | 0.688                 | 0.686              | 0.162               | 0.286              | 0.64                                  | 0.666                                 | 0.554                                 | 0.598                                 |
| Qwen/Qwen2.5-7B                      | 0.282                 | 0.504              | 0.128               | 0.144              | 0.452                                 | 0.462                                 | 0.39                                  | 0.424                                 |
| microsoft/Phi-3.5-mini-instruct      | 0.488                 | 0.49               | 0.312               | 0.352              | 0.55                                  | 0.57                                  | 0.462                                 | 0.468                                 |
| Llama-3.2-3B-Instruct                     | 0.522                 | 0.636              | 0.008               | 0.01               | 0.466                                 | 0.508                                 | 0.466                                 | 0.47                                  |
| Llama-3.2-1B-Instruct                     | 0.386                 | 0.398              | 0.01                | 0.016              | 0.312                                 | 0.336                                 | 0.286                                 | 0.312                                 |

---

### (2) Statistical Testing

We thank the reviewers for raising the concerns about the dataset size. We perform statistical testing to show that LongMemEval allows researchers to derive statistically significant results. Specifically, we used paired t-tests to verify that:

* Figure 4 (pilot study): in Figure 4a, offline reading is significantly better than ChatGPT or Coze with GPT-4o (p < 0.001) and in Figure 4b, the performance on LongMemEval_S is significantly worse than oracle for all the models (p < 0.001).
* Table 2 (key design): the best results in each column is significantly better than the second best (p < 0.001) except for end-to-end QA with L3.1-70B and session as the value granularity.
* Table 3 (key expansion): the best results in each column is significantly better than the second best (p < 0.001).
* Figure 7 (reading methods): JSON + CoN is significantly better than the second best method (p < 0.005).

---

Finally, we appreciate the detailed editorial comments from all the reviewers and have incorporated them into the updated PDF. Note that the table numbers above refer to the numbers of the original draft instead of the revised draft (which has one more table and one less figure in the main text).

Thank you,

Paper 9780 authors

---

### Comment · Area_Chair_iFqY · 2024-11-27

Dear reviewers,

Thank you for your efforts reviewing this paper. If you haven't, can you please check the authors' responses and see if your concerns have been addressed? Please acknowledge you have read their responses. Thank you!

---

### Author Response · Authors · 2024-11-28
**An Example**

To better illustrate the memory stages, we present an example of our memory system through a temporal reasoning question.

> ## Question Data

> ### Evidence Session 1 (six rounds in total; we show two rounds here for simplicity)

Date: 2023/03/08 (Wed)

Round 1.1:

User: I'm thinking of getting a new bike rack for my car. Do you have any recommendations? By the way, *I've been having some issues with my bike lately - in mid-February, I had to take it in for repairs because the gears were acting up and I couldn't shift properly*.

Assistant: New bike rack, eh? I'd be happy to help you with some recommendations! Before we dive into that, sorry to hear about the issues with your bike. …

Round 1.2:

User: I have a hybrid bike, and I want to carry just one bike. My car is a Toyota Corolla. I'm not sure about the type of mount, but I'd like something easy to install and remove. I'm open to different price ranges, but I'd like to know what I'm getting for my money.

Assistant: With a hybrid bike and a Toyota Corolla, you've got a great combo for commuting or casual rides. Based on your preferences, here are some bike rack options to consider:\n\n**Roof-Mounted Racks**\n\n* **Thule Evo 1-Bike Carrier** (around $100-$150): A popular, easy-to-install roof rack that's compatible with your Corolla. It's a great value for the price, and Thule is a trusted brand in the industry.\n* **Yakima FullBack 1-Bike Carrier** (around $150-$200): Another well-known brand, Yakima's FullBack is a sleek, low-profile rack that's easy to install and remove. It's a bit pricier than the Thule, but it's a great option if you want a more premium feel. …

> ### Evidence Session 2 (six rounds in total; we show two rounds here for simplicity)

Date: 2023/03/11 (Sat)

Round 2.1:

User: I'm thinking of getting a newer car and I'm researching the new hybrid model of my current Toyota Corolla. Do you have any info on its fuel efficiency and any promotions or discounts available? By the way, *I just washed my current Corolla on Monday, February 27th, after not doing so for over a month* - it was getting pretty dirty after that snowstorm we had a few weeks prior!

Assistant: Exciting to hear that you're considering upgrading to a newer hybrid model of your trusty Toyota Corolla!\n\nAccording to the latest data, the 2022 Toyota Corolla Hybrid has an estimated EPA rating of up to 53 mpg in the city, 52 mpg on the highway, and 52 mpg combined. …

Round 2.2:

User: I'd like to know more about the maintenance costs of the Corolla Hybrid. I've been doing some maintenance on my current car myself, like washing and vacuuming, but I had to take it to the dealership for a routine checkup this Friday because my friend said the "check engine" light came on when she borrowed it.

Assistant: Wise decision to get the "check engine" light checked out!\n\nMaintenance costs are an essential factor to consider when buying a new car, and the Toyota Corolla Hybrid is known for being relatively low-maintenance. Here are some points to keep in mind: …

---

Question: Which vehicle did I take care of first in February, the bike or the car?
Answer: Bike

Note that this question is only one type of temporal reasoning, where the dates are explicitly mentioned. For a lot of other questions, the dates of events are implied in the timestamps.

Now, we demonstrate a memory system with the proposed optimizations.

> ## Indexing

We use round as the value granularity. There are thus four values: round 1.1, 1.2, 2.1, and 2.2 among other rounds in the chat history. We additionally extract the following user facts, optionally with timestamps:

* Fact 1: (mid-February) The user took their bike for a repair in mid-February.

* Fact 2: (3/8) The user plans to buy a bike rack.

* Fact 3: (3/8) The user has a hybrid bike.

* Fact 4: (3/8) The user has a Toyota Corolla.

* Fact 5: (3/11) The user is thinking of replacing their current Toyota Corolla.

* Fact 6: (2/27) The user washed their Corolla on 2/27.

* Fact 7: (3/10) The user took their Toyota Corolla to the dealership for a routine checkup.

Using these facts for key expansion, the final index will have the following key-value structure:

KV1: (Fact 1, Round 1.1) -> Round 1.1

KV2: (Fact 2, Round 1.1) -> Round 1.1

KV3: (Fact 3, Round 1.2) -> Round 1.2

KV4: (Fact 4, Round 1.2) -> Round 1.2

KV5: (Fact 5, Round 2.1) -> Round 2.1

KV6: (Fact 6, Round 2.1) -> Round 2.1

KV7: (Fact 7, Round 2.2) -> Round 2.2

> ## Retrieval

For retrieval, in the query-based filtering stage, based on the question, a timestamp range 2/1 - 2/28 is first extracted, which is used to filter out KV2, KV3, KV4, KV5, and KV7, as they fall outside of the time range. Then, query-key similarity is used to retrieve most relevant items from the filter subsets.

> ## Reading

Finally, we provide the retrieved memory values (in this case, rounds) to the model with CoN as the reading strategy. A prompt example is presented in Figure 13.

---

### Author Response · Authors · 2024-12-01
**Looking forward to further discussion**

Dear Reviewers,

We hope this message finds you well. We want to gently remind you that the deadline for the discussion period is approaching. If there are any remaining points you would like us to address, we would be grateful for the opportunity to respond. We are more than happy to continue the conversation.

Best regards,

All authors

---

### Meta-Review · Area_Chair_iFqY · 2024-12-22

**Metareview:**

Summary:

Long-term memory capabilities of LLM-based chat assistants in sustained interactions remain underexplored. This paper introduces LongMemEval, a new benchmark designed to evaluate five core long-term memory abilities: information extraction, multi-session reasoning, temporal reasoning, knowledge updates, and abstention. LongMemEval consists of 500 meticulously curated questions embedded within freely scalable user-assistant chat histories, and presents a significant challenge to existing long-term memory systems.  Commercial chat assistants and long-context LLMs show a 30% accuracy drop on memorizing information across sustained interactions.

The paper also presents a unified framework that breaks down the long-term memory design into four design choices across the indexing, retrieval, and reading stages. Experiment results show that these designs greatly improve both memory recall and downstream question answering on LongMemEval.

Strengths:

1. The proposed benchmark is a new and challenging benchmark, which is valuable for long-term memory evaluation.

2. The paper identifies several core memory capabilities and formulates a unified framework.

3. The paper is promising to provide valuable resources and guidance for advancing the long-term memory capabilities of LLM-based chat assistants.

Weaknesses:

Most concerns or weaknesses raised in original reviews are addressed during the rebuttal. Here are some outstanding issues, solving which could further enhance the paper:

1. Deeper insights about the five aspects (information extraction, multi-session reasoning, temporal reasoning, knowledge updates, and abstention) for evaluating long-term memory should be discussed. (Reviewer eitt). The authors did provide a response that partly addresses this issue.

2. There seem to be many writing issues in the original draft and still some in the revised version (Reviewer LhuP).

**Additional Comments On Reviewer Discussion:**

Comments and concerns that have been addressed during the discussion period:

1. Reviewer NGvs raised concerns about only using GPT-4o and Llama 3.1-8B-instruct models for experiments. The authors provided further clarification and more experiments using more models as the reader model. They also listed as one weakness the need of lots of human intervention to make sure of the dataset quality, which I agree with the authors is more of an advantage than a limitation.

2. Reviewer eitt raised questions about the hallucination control when talking to AI chatbots and Reviewer GckN also had similar questions regarding chat history quality and timestamp quality. I think the authors have provided a satisfactory response to them.

---

### Decision · Program_Chairs · 2025-01-22

Accept (Poster)